# From Decoding to Meta-Generation:
# Inference-time Algorithms for Large Language Models

**Sean Welleck**                                                    *wellecks@cmu.edu*
*Carnegie Mellon University*

**Amanda Bertsch**[*]                                              *abertsch@cs.cmu.edu*
*Carnegie Mellon University*

**Matthew Finlayson**[*]                                           *mfinlays@usc.edu*
*University of Southern California*

**Hailey Schoelkopf**[*]                                           *hailey@eleuther.ai*
*EleutherAI*

**Alex Xie**                                                       *alexx@cs.cmu.edu*
*Carnegie Mellon University*

**Graham Neubig**                                                  *gneubig@cmu.edu*
*Carnegie Mellon University*

**Ilia Kulikov**                                                   *kulikov@meta.com*
*Meta FAIR*

**Zaid Harchaoui**                                                 *zaid@uw.edu*
*University of Washington*

*\*Co-second authors*

**Reviewed on OpenReview:** `https://openreview.net/forum?id=eskQMcIbMS`

## Abstract

One of the most striking findings in modern research on large language models (LLMs) is that scaling up compute during training leads to better results. However, less attention has been given to the benefits of scaling compute during inference. This survey focuses on these inference-time approaches. We explore three areas under a unified mathematical formalism: token-level generation algorithms, meta-generation algorithms, and efficient generation. Token-level generation algorithms, often called decoding algorithms, operate by sampling a single token at a time or constructing a token-level search space and then selecting an output. These methods typically assume access to a language model's logits, next-token distributions, or probability scores. Meta-generation algorithms work on partial or full sequences, incorporating domain knowledge, enabling backtracking, and integrating external information. Efficient generation methods aim to reduce token costs and improve the speed of generation. Our survey unifies perspectives from three research communities: traditional natural language processing, modern LLMs, and machine learning systems.

# Contents

Meta-generator

1. Generation algorithms
   - Maximization
   - Sampling
   - Controlled generation
2. Meta-generation
   - Programmatic patterns
   - External information
     - Multiple models
     - Tools
     - Environments
3. Efficient generation
   - Optimizing token cost
   - Speeding up generators
   - Speeding up meta-generators

Generator

```python
def generate_proof(llm, theorem):
    strategies = [
      "Prove by contradiction.\n",
      "Prove by induction.\n",
    ]
    candidates = [
        llm.generate(strategy + theorem)
        for strategy in strategies
        for sample in range(5)
    ]
    output = llm.generate(
        "Which of the proofs is best?\n"
        + "\n".join(candidates)
    )
    return output
```

Figure 1: Generation algorithms produce output text using a language model. Meta-generation algorithms are programs that interleave calls to generation algorithms with control flow and external information, yielding text. Our survey covers generation algorithms and their goals (§3), meta-generation patterns (§4) and sources of external information (§5), and efficiency in terms of token cost (§6) and speed (§7).

# 1 Introduction

One of the most striking findings in modern research on large language models (LLMs) is that, given a model and dataset of sufficient scale, scaling up the compute used at training time leads to better final results (Kaplan et al., 2020; Hoffmann et al., 2022). However, there is another, lesser-mentioned scaling phenomenon, where adopting more sophisticated methods or scaling compute at *inference time* (Jones, 2021) can result in substantially better outputs from LLMs. This survey focuses on these approaches by exploring three connected themes: token-level generation algorithms, meta-generation algorithms, and efficient generation.

*Token-level generation algorithms*, often called decoding algorithms, have a rich history in natural language processing, ranging from classical greedy decoding and beam search to modern sampling algorithms such as nucleus (Holtzman et al., 2020) and $\eta$-sampling (Hewitt et al., 2022). These methods operate by sampling one token at a time or constructing a token-level search space. They assume varying levels of access to a language model's internals, such as logits, next-token distributions, or probability scores.

Recently there has been growing interest in *meta-generation algorithms*—algorithms that operate on partial or full sequences, and treat the LLM as a black box that is called as part of a larger generation program (Figure 1; Khattab et al. (2022); Dohan et al. (2022); Schlag et al. (2023)). For example, a meta-generation algorithm for solving a math problem might generate multiple solution paths, evaluate the solutions with a calculator, then select the most common answer. Meta-generators can increase the compute resources devoted to generation by making multiple model calls, augmenting the model with search algorithms (Yao et al., 2023; Madaan et al., 2023), or incorporating external data sources. Doing so has seen success in improving task performance (e.g., problem solving (Lewkowycz et al., 2022)) and steering the output distribution (e.g., with human preferences (Stiennon et al., 2020)), and may offer a way to overcome limitations of standard LLMs such as error accumulation (Dziri et al., 2023) and computational capacity (Merrill & Sabharwal, 2024). Moreover, meta-generation research is widely accessible, as it often only requires black-box LLM access.

Finally, generation needs to be fast and cost-effective. Fast generation becomes increasingly challenging as models grow in size, while cost becomes a critical factor in meta-generation algorithms that call models many times. On the other hand, meta-generation algorithms open new kinds of shared computation that can be leveraged for improved efficiency. As a result, there is growing interest in *efficient generation algorithms* that speed up generation and reduce token costs by drawing on ideas from machine learning systems and

related areas. Efficient generation in turn expands the frontier of algorithms that are feasible to experiment with and develop, leading to a virtuous cycle of algorithmic development.

Our survey provides a unified treatment of these three themes: token-level generation algorithms, meta-generation algorithms, and techniques for making generation fast and cost-effective. We integrate ideas from traditional natural language processing, modern LLMs, and machine learning systems, and present a mathematical formalism that includes both classical generation algorithms and modern meta-generators. This unified view is particularly important as the field expands. For example, practitioners working on novel meta-generation algorithms may benefit from learning about the historical context of generation algorithms or practical efficiency constraints, while researchers interested in efficiency may benefit from learning about major algorithmic patterns. More broadly, we aim to promote further research on inference-time approaches.

**Comparison to existing surveys.** Several prior surveys have focused on training-time methods for better text generation (Li et al., 2021; Lu et al., 2018). Wiher et al. (2022) presents a detailed analysis of a smaller set of decoding strategies, while Zarrieß et al. (2021) spotlight token-level methods, with a particular focus on considerations for encoder-decoder models. In parallel, several surveys have addressed prompting and related methods (Liu et al., 2021b; Sahoo et al., 2024), though these works do not address token-level methods. Recent surveys have also considered strategies for speeding up inference (Chitty-Venkata et al., 2023; Miao et al., 2023a; Khoshnoodi et al., 2024; Wang et al., 2024a). However, these works focus primarily on token-level generation, not meta-generation; as a result, the discussion of inference-time compute-performance tradeoffs is limited. Our survey unifies and draws connections across these three areas. Finally, Xiao et al. (2023) focus on non-autoregressive generation, while our survey focuses on autoregressive generation.

**Roadmap.** This paper provides a survey of algorithms for token-level generation, meta-generation, and efficient generation, summarized in Figure 1. First, we consider why we use generation algorithms at all. Generally, a user's intent is to surface a high-quality output from the model, which we formalize and discuss in §2. Readers who would like to review terminology or follow the mathematical formulation of the survey in depth should start in this section. Next, we discuss token-level generation algorithms in detail in §3. Most algorithms referred to as "decoding algorithms" in the literature are covered in this section. We discuss these methods' theoretical motivation, practical impact, commonalities, and provide a unified frame for discussion. These methods generally require some degree of access to the model's internals.

A growing set of methods operate over partial or full sequences rather than individual tokens. These *meta-generation* algorithms have emerged from several communitites, including researchers interested in designing new decoding algorithms or prompting methods, as well as researchers interested in language model alignment and reasoning. Works from these communities often have different motivations and use different terminology. We present a unified picture in §4, classifying them according to their *programmatic structure* (e.g., parallel generation, search, or refinement), and discussing their motivations.

In addition to wanting a high-quality output, we often care about the *efficiency* of generation. We consider two definitions of efficient generation. In §6 we consider the token cost of generation algorithms, which is especially relevant for studying cost-performance tradeoffs as the amount of computation allocated to generation is scaled up, and for those using API-access models that charge by the token. In §7, we discuss methods for speeding up generation primarily from a systems perspective, where access to the model weights is assumed and latency and throughput are the key considerations. In this section, we draw upon work primarily from the machine learning systems (MLSys) community. The section serves as both an introduction to this area for machine learning researchers whose work does not focus on systems, and a practical exploration of tools for speeding up generation. We include a review of libraries that implement the described techniques.

We conclude the survey by discussing takeaways, broader directions, and future work in §8.

## 2 Preliminaries

Generation algorithms are used to produce outputs from a trained language model. Language models are probabilistic models over sequences, $p_\theta(y|x)$, and most generation algorithms attempt to either find highly probable sequences or sample from the model's distribution. A natural question is *why are sophisticated*

| Symbol | Name | Explanation/example |
|---|---|---|
| $p_\theta$ | Model distribution | The conditional distribution defined by an LM with parameters $\theta$ |
| $s_\theta$ | Model logit function | Assigns LM scores to tokens. Normalizing gives $p_\theta$. |
| $p_*$ | Training distribution | The target distribution for which an LM was trained. |
| $A$ | Acceptability function | Takes a set $S$ of outputs, assigns an acceptability score. |
| $r$ | Reward function | Proxy for $A$. |
| $v$ | Scoring function | Proxy for $r$. Also: *value function, learned verifier, reward model.* |
| $g$ | A generator | E.g., a sampling algorithm for an LM, or a refinement algorithm. |
| $g(\cdot|x)$ | Generator distribution | The distribution obtained by applying a generator to an input $x$. |
| $q_*$ | Target distribution | E.g., the distribution of utterances in the style of a helpful assistant. |
| $f$ | Deterministic function | E.g., a deterministic generator such as greedy decoding, or a calculator. |
| $\phi$ | A generator's parameters | E.g., temperature, number of samples. |
| $d$ | Distance function | Measures distance, e.g., KL divergence, between two distributions. |
| $\mathcal{Y}$ | Output space | The set of possible LM outputs, i.e., strings. |
| $\mathcal{V}$ | Vocabulary | The set of tokens that make up sequences in $\mathcal{Y}$. |
| $\mathcal{P}(\mathcal{Y})$ | Probability distributions | The set of probability distributions over outputs. |
| $y_t$ | Current token | The token at index $t$ in a sequence $y \in \mathcal{Y}$. |
| $y_{<t}$ | Prefix, or context | The tokens preceding index $t$ in a sequence $y \in \mathcal{Y}$. |

Table 1: An overview of symbols for convenience.

*generation algorithms needed at all?* For example, we might imagine that simply sampling once from the model's unmodified output distribution, $y \sim p_\theta(y|x)$ is sufficient. We begin by defining some terminology (summarized in Table 1), and then present general goals of generation which shed some light on this question.

## 2.1 The user's goal in generation

When a user is generating outputs with a language model, it may be with one or more goals in mind. The user may want output that is as high quality as possible for some notion of quality, such as a correct answer to a math problem or a factual and well-written summary. The user may want multiple outputs, such as alternative solutions to a problem or multiple summaries to read through and synthesize. In general, users now access language models through general-purpose text-in text-out APIs, making it impossible to enumerate all of the specific use cases or goals that a user might have.

As a result, to formalize an overall goal for generation, we will need to take a fairly general perspective. We assume that the user has some underlying measure of "acceptability" for any set $S$ of outputs, $A(S) \in \mathbb{R}$. For example, a single sequence set may have high acceptability if it represents a correct solution to a problem, while in a different context a set $S$ may have high acceptability if it balances some notion of diversity with some notion of quality. The acceptability scores, when normalized, form a probability distribution that we call the *target distribution $q_*$*,

$$q_*(S) \propto A(S). \tag{1}$$

Next, we treat generating outputs with a language model as sampling from a *generator $S \sim g$* that produces a set of sequences each time it is called. Finally, we assume that a user wants the distribution of outputs from the generator to be "close" to the distribution of their acceptability scores according to some proximity measurement $d$ between distributions. An ideal generator $g$ would thus satisfy:

$$\arg\min_g d(q_*, g). \tag{2}$$

In practice, we typically do not know how to measure the user's acceptability nor their desired notion of proximity, let alone how to design a generator that is guaranteed to produce outputs with high acceptability. At a high level, the remainder of this survey can be seen as surveying ways to design generators that optimize some proxy of acceptability in an efficient way. For example, some algorithms will try to produce a single output that is acceptable with a language model's probability as a proxy of acceptability. Other algorithms

will try to directly sample from some target distribution that we may interpret as being a proxy to a user's target distribution. To begin with, let us go into more detail on what a "generator" is, starting with the definition of a language model, a generation model, and a generation algorithm.

## 2.2 The modeling problem

**Language models.** Let $p_\theta$ be a language model that approximates the distribution $p_*$, denoted $p_\theta \approx p_*$. We consider autoregressive language models $p_\theta(y|x) = \prod_{t=1}^{T} p_\theta(y_t|y_{<t}, x)$, where $y$ is a sequence of tokens from vocabulary $\mathcal{V}$. Each conditional distribution is of the form, $p_\theta(\cdot|y_{<t}, x) = \exp(s_\theta(\cdot|y_{<t}, x))/Z$, where $s_\theta(\cdot|y_{<t}, x) \in \mathbb{R}^{|\mathcal{V}|}$ are referred to as logits and $Z = \sum_{i=1}^{|\mathcal{V}|} \exp(s_\theta(\cdot|y_{<t}, x))_i$. We henceforth refer to a model of this form as simply a language model (LM) for brevity.

A **generation model associated with a language model** $p_\theta$ is a function $g : \mathcal{X} \times \Theta \times \Phi \to \mathcal{P}(\mathcal{Y})$ that maps an input $x \in \mathcal{X}$, a model $p_\theta$ with $\theta \in \Theta$, and any additional parameters $\phi \in \Phi$ to a probability distribution over outputs, $g(y|x; p_\theta, \phi) \in \mathcal{P}(\mathcal{Y})$.

Calculating the probability distribution over outputs $g(y|x; p_\theta, \phi) \in \mathcal{P}(\mathcal{Y})$ is in most situations analytically intractable. One can use the **generation algorithm** in order to obtain independent or dependent samples from $g(y|x; p_\theta, \phi) \in \mathcal{P}(\mathcal{Y})$; we refer to this process as **generating**, $y \sim g(y|x; p_\theta, \phi)$). We will also refer to $g$ as a **generator**, and the distribution obtained by applying $g$ to an input as a **generation distribution**. Generation algorithms may be deterministic or stochastic. We denote generating from a **deterministic generator** as $y = f(x; p_\theta, \phi)$, where $f$ is the deterministic generator. While methods to maximize a scoring function are often deterministic and methods for generating sets of outputs are often stochastic, in practice each kind of method can be used toward either goal.

Let us now return to the general goal of generation that we formulated above. For notational simplicity, let us consider generating a single sequence, i.e. $S = \{y\}$. In practice, we can design a generation algorithm to maximize some proxy $r(\cdot)$ of acceptability:

$$\arg\max_{g} r\left(q_*(\cdot|x), g(\cdot|x; p_\theta, \phi)\right), \tag{3}$$

where $r : \mathcal{P}(\mathcal{Y}) \times \mathcal{P}(\mathcal{Y}) \to \mathbb{R}$ is some reward function between distributions. We group generation methods into 3 categories: methods for maximization (§2.2.1), sampling from the model (§2.2.2), and sampling from a target distribution (§2.2.3). These are special cases of (3). For maximization, $q_* \propto v$ for a scoring function $v$, and we have $r(q_*, g) = \mathbb{E}_{y \sim g} q_*(y|x)$. For sampling from a target distribution, $r$ is a divergence between the target distribution and the generation distribution.

### 2.2.1 Maximization

We define *decoding* as the process of maximizing a score either deterministically or with a high probability:

**Definition 1** (Score maximizing algorithm). A score maximizing algorithm for score $v : \mathcal{S} \to \mathbb{R}$ refers to an algorithm that approximates:

$$f(x; p_\theta, \phi) = \arg\max_{S \in P(\mathcal{Y})} v(S), \tag{4}$$

where $P(\mathcal{Y})$ is the set of all subsets of $\mathcal{Y}$. Decoding algorithms can be greedy algorithms, concave maximization algorithms, combinatorial optimization algorithms, or stochastic algorithms.

### 2.2.2 Sampling

Some algorithms are designed to sample from a distribution. The samples can then be used for any purpose, such as for maximizing some task-specific metric (e.g., a metric that balances quality and diversity).

**Definition 2** (Sampler for $q(p_\theta)$). A sampler for $q(p_\theta)$ gives a sample from some distribution $q_p$ proportional to $q(p_\theta)$, where $q$ is a map between probability distributions:

$$y \sim q_p \propto q(p_\theta(y|x)). \tag{5}$$

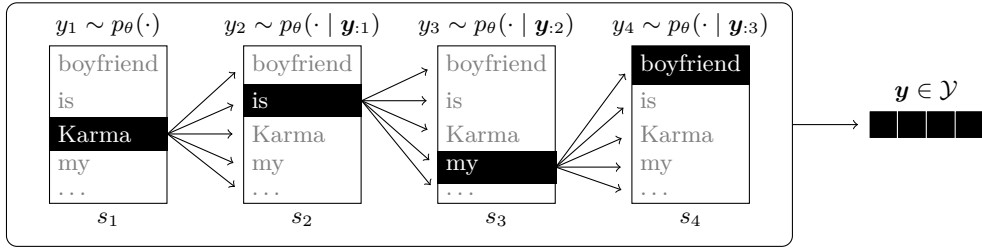

Figure 2: Sampling algorithms choose the next token at each time step $s_i$ by sampling from the conditional distribution $p_\theta(\cdot \mid \boldsymbol{y}_{:i-1})$ and appending it to the context.

### 2.2.3 Sampling from a specified target distribution

In some cases we can specify which target distribution $q_*$ an algorithm is aiming to sample from.

**Definition 3** (Sampler for a target distribution). A generator $g$ is called a sampler for target distribution $q_*$ if it approximates:

$$\max_g -\mathrm{D}_{\mathrm{KL}}\left(q_*(\cdot|x)\|g(\cdot|x;p_\theta,\phi)\right).^1 \tag{6}$$

An optimal sampling algorithm for $q_*$ yields an unbiased sample $y \sim q_*(\cdot|x)$.

Next, we will see examples of algorithms that achieve these goals by generating token-by-token.

## 3 Token-level generation algorithms

In this section, we discuss representative methods that operate on the token-level (e.g., by sampling a single token at a time, or constructing a token-level search space and then selecting an output at the end).

Methods in this category generally assume access to a language model's logits, next-token distributions, or probability scores. These methods will later be treated as black boxes that are called by meta-generators.

### 3.1 MAP decoding algorithms

When faced with the question of what sequence to choose from the distribution defined by a language model, a natural objective is to choose the *most likely sequence*. Several popular decoding algorithms therefore attempt to find a generation $y$ that maximizes $p_\theta(y|x)$, referred to as maximum a-posteriori (MAP) decoding.

**Definition 4** (MAP decoding algorithms). A MAP decoding algorithm approximates:

$$f(x;p_\theta,\phi) = \arg\max_{y\in\mathcal{Y}} p_\theta(y|x). \tag{7}$$

The term MAP comes from viewing $p_\theta$ as a posterior over outputs $y$ given the observed input $x$, and *decoding* comes from information theory.

**Greedy decoding.** Arguably the simplest MAP decoding algorithm is *greedy decoding*, which generates a sequence $\hat{y}_1, \ldots, \hat{y}_T$ by recursively selecting the highest probability token from the next-token distribution:

$$\hat{y}_t = \arg\max_{y_t\in\mathcal{V}} p_\theta(y_t|\hat{y}_{<t},x), \tag{8}$$

for $t = 1, \ldots, T$, with $T$ determined by a stopping condition (e.g., a fixed $T$ or $y$ including a certain string). Greedy decoding is an *approximate* MAP decoding algorithm, meaning that it finds a sequence that is not

---

[1]Any divergence $d$ with the property that $q_* = q$ iff $d = 0$ is suitable.

necessarily a maximizer of (7). Specifically, it approximates (7) as:

$$\arg\max_{y \in \mathcal{Y}} p_\theta(y|x) = \arg\max_{(y_1,\dots,y_T) \in \mathcal{Y}} \prod_{t=1}^{T} p_\theta(y_t|y_{<t}, x) \tag{9}$$

$$\approx \left( \hat{y}_1 = \arg\max_{y_1 \in \mathcal{V}} p_\theta(y_1|x), \cdots, \hat{y}_T = \arg\max_{y_T \in \mathcal{V}} p_\theta(y_T|\hat{y}_{<T}, x) \right). \tag{10}$$

Despite its naive approximation, greedy decoding is a widely-used generation algorithm. For instance, it is used in Google's Gemini report (Gemini Team et al., 2023), and is available on typical language model APIs.

**Other MAP decoding algorithms.** Several algorithms have been designed that typically return better approximations (i.e., more probable sequences) than greedy decoding. In the context of neural sequence-to-sequence models, *beam search* (Graves, 2012; Sutskever et al., 2014) is a widely-studied MAP decoding algorithm. It maintains a data structure of multiple prefixes $y_{<t}$ at each generation step, expands each prefix with each possible next-token, $y_{<t} \circ y_t$, scores each expanded prefix with $p_\theta(y_{<t} \circ y_t|x)$, and retains the top-$K$ expanded prefixes for the next iteration. This can be seen as a generalization of greedy decoding, which expands only a single prefix. In practice, beam search has been shown to improve upon greedy decoding in terms of downstream task performance in many settings (e.g., Sutskever et al. (2014); Freitag & Al-Onaizan (2017); Kulikov et al. (2019)). It has several variations and generalizations that we will return to when we take the perspective of generation as search (§4.3). Although the space of possible outputs is extremely large, it is sometimes possible to find an *exact* MAP solution (i.e., a sequence that maximizes (7)). For instance, Stahlberg & Byrne (2019) combine elements of beam search and depth-first search to perform exact search with machine translation models, which was improved upon in Stahlberg et al. (2022).

**Pitfalls of MAP decoding.** Despite its popularity, several studies suggest that the MAP decoding objective is not desirable (Meister et al., 2020). Empirically, MAP decoding has a tendency to produce degenerate results. For example, Koehn & Knowles (2017) found that wide beam search (which approaches exact MAP decoding in the limit) degrades neural machine translation (NMT) outputs by favoring shorter outputs. In fact, Stahlberg & Byrne (2019) found that exact MAP decoding often returned the *empty sequence* in NMT. Length normalization (e.g., dividing the log-probability of the sequence by its length) can mitigate MAP decoding's tendency to favor shorter sequences (see Murray & Chiang, 2018), but this is only a heuristic and does not fully counteract degradation for the largest beam sizes (Koehn & Knowles, 2017). Approximate MAP decoding, e.g., greedy, can also fail by getting trapped in repetitive sequences (Holtzman et al., 2020; Welleck et al., 2020; Eikema & Aziz, 2020).

There are several explanations for degenerate behavior in MAP decoding, a phenomenon known as the *inadequacy of the mode* (Eikema, 2024). Some studies attribute degenerative phenomena in MAP decoding to the tendency of the most likely generations to accumulate so little probability that the mode becomes arbitrary due to small errors in probability estimation (Eikema & Aziz, 2020; Stahlberg et al., 2022). In an alternative explanation, Meister et al. (2023b) use information-theoretic analysis to show that MAP decoding generations often fall outside of the *typical set* of sequences in the language model's distribution. To illustrate how this occurs, consider that the most probable outcome of 100 flips of a slightly biased coin (with 0.51 probability of heads, 0.49 probability of tails) is a sequence of 100 heads. However, this result would be atypical; a close-to-even mix of heads and tails would be more typical (Dieleman, 2020).

**Unreasonable effectiveness of approximate MAP decoding.** Despite the drawbacks of MAP decoding, rough approximations of MAP decoding remain popular in the forms of greedy decoding and narrow beam search. For example, Shi et al. (2024a) study Llama 2 language models and find that beam search performs well on input-output tasks such as translation. Meister et al. (2020) hypothesize that these decoding methods are effective because they inadvertently enforce information-theoretic patterns that are characteristic of human text. On the other hand, the MAP objective can limit diversity, and beam search in particular can incur a large computational cost. Increasingly, practitioners using an inference library or language model API may instead turn to algorithms that efficiently sample from the model,[2] which we describe next.

---

[2]For example, the widely-used Open AI API does not support beam search.

### 3.2 Sampling and adapters

A popular alternative to the MAP objective is to sample directly from the language model's distribution $y \sim p_\theta(y|x)$.

**Ancestral sampling.** The most basic sampling algorithm for $p_\theta$ is motivated by the fact that autoregressive models decompose sequence probabilities into a product of next-token conditionals:

$$p_\theta(y|x) = \prod_{t=1}^{|y|} p_\theta(y_t|y_{<t}, x), \tag{11}$$

where $y = (y_1, \ldots, y_{|y|})$ and $y_t$ are individual tokens. As shown in Figure 2, sampling from this model can be done recursively,

$$y_t \sim p_\theta(\cdot|y_{<t}, x), \tag{12}$$

where $y_0$ is a given starting token, and the algorithm terminates upon reaching a particular token or a given length. The result is mathematically equivalent to sampling a sequence $y$ directly from $p_\theta(\cdot \mid x)$, and is known as *ancestral sampling.* Other algorithms such as speculative sampling (Leviathan et al., 2022) aim to sample from $p_\theta$ more efficiently, which we will discuss in more detail later in the review (§7).

**Sampling, MAP, and the diversity-coherence trade-off.** Ancestral sampling avoids many of the degenerate behaviors of MAP decoding, such as repetition traps, and introduces more diversity into LM generations. However, ancestral sampling can suffer from *incoherence*, i.e., over-sampling highly-unlikely tokens due to model error (Zhang et al., 2021). Hewitt et al. (2022) hypothesize that this occurs because perplexity-based loss functions encourage language models to over-estimate the probability of unlikely tokens to avoid large loss penalties (a behavior called mode-seeking). Alternatively, Finlayson et al. (2024a) hypothesize that constraints imposed by the LM's output layer, i.e., the softmax bottleneck (Yang et al., 2018), cause model errors, and propose a method, basis-aware truncation (BAT), to avoid these errors.

**Balancing the diversity-coherence tradeoff.** Several decoding strategies attempt to balance the diversity-coherence tradeoff by interpolating between greedy and ancestral sampling. These include nucleus (Holtzman et al., 2020), top-$k$ (Fan et al., 2018), and $\eta$- and $\epsilon$-sampling (Hewitt et al., 2022), which use various heuristics to choose a threshold at each time step and only sample tokens with probability greater than the threshold. Another approach, temperature sampling (Ackley et al., 1985; Hinton et al., 2015), scales the LM logits to interpolate between greedy sampling and uniform sampling (setting all token probabilities equal), which can be useful when one wants *more* diversity than ancestral sampling offers.

### 3.3 Token-level sampling adapters

Except for beam search, all of the token-level sampling methods discussed so far can be viewed as *sampling adapters* $q_t$ (Meister et al., 2023a) which adjust each next-token distribution,

$$y_t \sim q_t(p_\theta(y_t|y_{<t}, x)). \tag{13}$$

**Example 1** (Temperature sampling as an adapter)**.** Temperature sampling adjusts the distribution by dividing the logits by a scalar *temperature* parameter $\tau$:

$$q_t(y_t|y_{<t}, x; p_\theta, \tau) \propto \exp\left(s_\theta(y_{<t}, x)/\tau\right). \tag{14}$$

Sending $\tau$ to 0 yields greedy decoding, $\tau = 1$ yields ancestral sampling, and $\tau > 1$ approaches uniform sampling (all tokens have the same probability).

Many other token-level decoding methods can be cast as sampling adapters, including methods that re-weight logits with outputs from another model (Liu et al., 2021a; Li et al., 2023a), and a variety of other transformations summarized in Table 2. Many of these token-level generation algorithms assume access to

| Method | Purpose | Adapter | Extrinsic |
|---|---|---|---|
| Ancestral sampling | $y \sim p_\theta$ | – | – |
| Temperature sampling [1] | $y \sim q(p_\theta)$ | Rescale | – |
| Greedy decoding | $y \leftarrow \max p_\theta$ | Argmax (temperature$\rightarrow 0$) | – |
| Top-k sampling [56] | $y \sim q(p_\theta)$ | Truncation (top-k) | – |
| Nucleus sampling [86] | $y \sim q(p_\theta)$ | Truncation (cumulative prob.) | – |
| Typical sampling [154] | $y \sim q(p_\theta)$ | Truncation (entropy) | – |
| Epsilon sampling [82] | $y \sim q(p_\theta)$ | Truncation (probability) | – |
| $\eta$ sampling [82] | $y \sim q(p_\theta)$ | Truncation (prob. and entropy) | – |
| Mirostat decoding [11] | Target perplexity | Truncation (adaptive top-k) | – |
| Basis-aware sampling [57] | $y \sim q(p_\theta)$ | Truncation (linear program) | LP Solver |
| Contrastive decoding [129] | $y \sim q(p_\theta)$ | $\log p_{\theta'} - \log p_\theta$ and truncation | Model $p_{\theta'}$ |
| DExperts [137] | $y \sim q_*(\cdot\|x,c)$ | $\propto p_\theta \cdot (p_{\theta+}/p_{\theta-})^\alpha$ | Models $p_{\theta+}, p_{\theta-}$ |
| Inference-time adapters [146] | $y \sim q_* \propto r(y)$ | $\propto (p_\theta \cdot p_{\theta'})^\alpha$ | Model $p_{\theta'}$ |
| Proxy tuning [138] | $y \sim q_*(\cdot\|x,c)$ | $\propto p_\theta \cdot (p_{\theta+}/p_{\theta-})^\alpha$ | Models $p_{\theta+}, p_{\theta-}$ |

Table 2: Survey of token-level generation. $r(y)$ is a scalar reward function. $c$ is a control attribute. Extrinsic refers to a model or solver separate from the underlying language model $p_\theta$.

the language model's next-token distributions. In practice, next-token distributions are increasingly not provided by common generation APIs, both for practical reasons and for security (Finlayson et al., 2024b; Carlini et al., 2024). Instead, token-level algorithms are often implemented by the API provider, and used by setting hyperparameters (e.g., setting a temperature $\tau$).

**Adapters for statistical control.** Several decoding methods use sampling adapters to control the statistical and information-theoretic properties of model outputs and align them with those of human text. These include locally typical sampling (Meister et al., 2023b), which aims to sample from the LM distribution's typical set (MacKay, 2004); and mirostat sampling (Basu et al., 2021), which attempts to match the perplexity of the generated text to the expected perplexity under Zipf's law (Zipf, 1999; Powers, 1998). Intriguingly, Shi et al. (2024a) evaluate Llama 2 models with a variety of adapters (temperature, top-$k$, top-$p$, $\eta$, Mirostat, and typical sampling), and find no definitive best method for the evaluated open-ended text generation tasks. Furthermore, temperature sampling usually outperformed the other adapters in input-output tasks such as code generation and translation. In general, which adapter to use remains an open question.

**Autoregression and lookahead adapters.** Token-level algorithms generate from left-to-right, meaning that they generate each token without knowing the eventual identity of tokens to the right. Several algorithms have incorporated various heuristic scores $v(y_{\leq t})$ that adjust the next-token distribution using information from potential *future* tokens. This includes explicitly generating several tokens ahead (e.g., Lu et al. (2022); Leviathan et al. (2022)), or learning a function $v_\phi(y_{\leq t})$ that predicts a property of a full sequence (e.g., its style score or correctness) (Yang & Klein, 2021). Doing so can aid in satisfying sequence-level criteria.

**Distribution adjustment with another language model.** Some algorithms adjust the next-token distribution using another language model. This can arise from several motivations, including removing abnormalities in the model's next-token distributions (Li et al., 2023a), speeding up generation (Leviathan et al., 2022), or shifting the generation distribution to one with a property (e.g., a style) (Liu et al., 2021a).

## 3.4 Controlled generation

Many scenarios can be framed as aiming to sample from a language model's distribution modulated by a sequence-level criterion $c(y)$ (Korbak et al., 2022a;c; Hu et al., 2024; Zhao et al., 2024a):

$$q_* \propto p_\theta(y|x)c(y). \tag{15}$$

For example, $c(y)$ may assign high values to sequences with a particular style, or low values to sequences with toxic content or buggy code. Another way of phrasing (15) is sampling from a particular energy-based model (LeCun et al., 2006; Khalifa et al., 2021). We discuss three examples based on the structure of $c(y)$.

**Classifier.** In some cases $c(y)$ is a classifier $p(a|x,y)$, which predicts the probability that $y$ contains an "attribute" $a$, such as a style or non-toxicity. The goal is then to sample from:

$$q_* \propto p_\theta(y|x)p(a|x,y)^\beta, \tag{16}$$

where $\beta$ is a hyperparameter assigning more weight to the classifier at higher values of $\beta$. Various generation algorithms have been developed for this purpose, such as approximations based on reweighting next-token distributions with other language models (Liu et al., 2021a), reweighting with a learned classifier that approximates the sequence-level classification $p_\phi(a|y_{<t},x) \approx p(a|y,x)$ (Yang & Klein, 2021), or additional training to sample from $q_*$ (Khalifa et al., 2021; Hu et al., 2024; Zhao et al., 2024a).

**Indicator.** A special case is $c(y)$ indicating whether $y$ falls into a target set $Y_x^*$, such as the set of correct solutions to a reasoning problem, or sequences that have desired keywords. The goal is then to sample from:

$$q_* \propto p_\theta(y|x)\mathbb{I}[y \in Y_x^*], \tag{17}$$

where $\mathbb{I}[y \in Y_x^*]$ is 0 when $y \in Y_x^*$ and 1 when $y \notin Y_x^*$. Various generation algorithms incorporate a learned verifier $v_\phi(x,y) \approx \mathbb{I}[y \in Y_x^*]$ to aid in achieving this goal (Cobbe et al., 2021; Lightman et al., 2024), or design beam search heuristics for the case of desired keywords (Hokamp & Liu, 2017; Lu et al., 2022).

There is a clear connection between sampling from a target distribution of the form (17) and maximizing a scoring function (§2.2.1): sampling from (17) maximizes $v(y) = \mathbb{I}[y \in Y_x^*]$, e.g., correctness.

**Reward.** An important case is when $c(y)$ is governed by a *reward function* $r(x,y) \to \mathbb{R}$:

$$q_* \propto p_\theta(y|x)\exp\left(\frac{1}{\beta}r(x,y)\right), \tag{18}$$

where $\beta \in \mathbb{R}$ interpolates between sampling from $p_\theta$ ($\beta \to \infty$) and maximizing reward ($\beta \to 0$).

A notable example is aligning the distribution of generated text with a distribution of text preferred by humans (Ziegler et al., 2020; Stiennon et al., 2020; Ouyang et al., 2022). One way of operationalizing this problem is as one of finding a policy $\pi$ that balances maximizing a reward $r(x,y)$ that quantifies human preferences with generating sequences that are probable under a pretrained model $p_\theta$:

$$\max_\pi \mathbb{E}_{x\sim p, y\sim p_\theta(y|x)}[r(x,y)] - \beta\mathrm{KL}\left(\pi(y|x)\|p_\theta(y|x)\right). \tag{19}$$

The policy that maximizes the above objective is (Korbak et al., 2022b; Rafailov et al., 2023):

$$q_*(y|x) = \frac{1}{Z(x)}p_\theta(y|x)\exp\left(\frac{1}{\beta}r(x,y)\right), \tag{20}$$

where $Z(x)$ is a normalization factor. One strategy for sampling from $q_*$ is updating $p_\theta$ with reinforcement learning, then using ancestral sampling. This strategy is referred to as reinforcement learning from human feedback (Askell et al., 2021). Later, we will discuss meta-generation algorithms for addressing this problem.

A second approach is to re-weight each next-token distribution during autoregressive sampling. For example, reward-augmented decoding (Deng & Raffel, 2023) assumes access to a reward function $r(y_{\leq t}, x)$ that assigns a scalar reward to partial generations $y_{\leq t}$. It re-weights the tokens with the top-$k$ next-token probabilities using the reward, and samples from the re-weighted distribution. That is,

$$y_t \sim \mathrm{softmax}\left(s_\theta^{1:k}(y_{<t},x) + \beta r^{1:k}\right), \tag{21}$$

where $s_\theta^{1:k}(y_{<t}, x) \in \mathbb{R}^k$ are the top-$k$ logits at timestep $t$, $r^{1:k} \in \mathbb{R}^k$ are the rewards evaluated after appending each of the top-$k$ tokens to the prefix $y_{<t}$, and $\beta \in \mathbb{R}$ is a hyper-parameter. Inference-time policy adapters (Lu et al., 2023) directly optimizes an "adapter" language model to adjust a base language model's next-token distributions so that the combined model's generations receive higher rewards. Specifically, an adapter language model $p_\phi$ is combined with a base language model to form a "tailored policy",

$$p(\cdot|y_{<t}, x) \propto p_\theta(\cdot|y_{<t}, x)p_\phi(\cdot|y_{<t}, x), \tag{22}$$

and the tailored policy is updated with reinforcement learning while freezing the base model $p_\theta$'s parameters.

## 3.5 Constrained decoding

Often it is desirable to generate tokens that satisfy some constraint. These constraints may be structural, e.g., satisfying a JSON schema; or lexical, i.e., containing specific words. In contrast to *controlled* generation, *constrained* generation mandates that the output conforms to specific, formal criteria. Several methods from controlled generation can be employed to encourage LMs to conform to constraints, but here we focus on methods that strictly enforce such constraints. Naïvely, one can strictly enforce constraints via rejection sampling (§4.2.3), though there is no guarantee that this method will terminate. In this section, we therefore focus on *efficient* methods for constrained decoding.

### 3.5.1 Parser-based decoding for structural constraints

Structural constraints often take the form of a grammar. A simple example of this would be to constrain the language model output to valid US phone numbers. Another would be to constrain outputs that conform to a specific JSON schema. Structural constrained decoding methods enforce such constraints by pairing the language model with a parser. The parser filters next-token candidates by allowing only valid continuations. The central challenges of parser-based constrained decoding are efficiently identifying valid next-tokens with the parser and interfacing the parser with a sub-word tokenizer.

**Formal languages and parser efficiency.** The efficiency of a parser at identifying valid next-tokens depends on the complexity of the formalism used to specify the set of valid outputs. *Regular expression* (RegEx)-based parsers; e.g., Guidance (Lundberg et al., 2024), LMQL (Beurer-Kellner et al., 2022), and Outlines (Willard & Louf, 2023); can be implemented as finite-state automata, meaning that these can recognize valid continuations in constant time and space. More complex constraints, such as JSON specifications, require more sophisticated, computationally expensive parsers. Though JSON is technically not a context-free grammar (CFG), several CFG-based parsers including PICARD (Scholak et al., 2021), GCD (Geng et al., 2023), Synchromesh (Poesia et al., 2022), and Domino (Beurer-Kellner et al., 2024) can be used to enforce valid JSON outputs.

**Template speedups** Parser-based constrained decoding methods can accelerate decoding, i.e., when there is only a single valid next token, that token can be immediately accepted without doing a forward pass with the model. Highly templated RegEx-based methods benefit most from this property (Lundberg et al., 2024).

**Token healing.** One drawback of constrained decoding is that constraints may cause the language model to exhibit unusual behaviors due to the template or grammar forcing an unnatural token boundary. Several decoding methods deal with this issue by employing a method called *token healing* (Lundberg, 2023). Token healing rolls back the tokenizer to the penultimate token then chooses the next token with the constraint that it must have the remaining characters as a prefix (see Figure 3 for an example). This method uses a prefix tree to track valid continuations.

**Minimally invasive constrained decoding.** Constrained LLM performance can also be limited by templates that are too inflexible. Over-constrained templates can prevent the model from generating outputs that would otherwise be valid. (Beurer-Kellner et al., 2024) call this property the *invasiveness* of a constrained decoding method, and refer to a method as being *minimally invasive* if every valid output that the model can generate can still be generated under constrained decoding. They link the invasiveness of

| Template | Without healing | With healing |
|---|---|---|
| "An unnatural token" | An unnatural token i zat ion | An unnatural tokenization |

Figure 3: Pre-tokenizing templates can cause issues for (greedy) tokenization by forcing the model to break tokens at unnatural points (e.g., at the end of the word "token"). Token healing rolls back the tokenizer by one token (back to "unnatural") then enforces that the continuation begin with the remaining text "token".

constrained decoding methods to explosions in output perplexity and attempt to mitigate these drawbacks with their method DOMINO.

### 3.5.2 Lexically constrained decoding

It is often desirable to constrain language model outputs to contain or not contain specific words. LLM inference APIs often allow users to specify token ban lists or logit bias to discourage the model from outputting specific words, but forcing LLMs to output specific words is more challenging. Lexically constrained decoding often employs search (Hokamp & Liu, 2017) to find likely generations that satisfy the constraints. Some methods seek to improve this search, such as through gradient guidance (Kumar et al., 2022).

For more complex lexical constraints, NEUROLOGIC decoding (Lu et al., 2021) allows users to specify constraints as a logical formula in conjunctive normal form (CNF) then enforces these constraints via a modified beam search, e.g., (food ∨ foods) ∧ (table ∨ tables) specifies that the generation must contain either "food" or "foods" and either "table" or "tables". Followup work (Lu et al., 2022) adds a lookahead to this approach.

In summary, we have seen several strategies for constructing a token-level search space and adjusting the next-token distributions of a model during sampling. Next, we will treat these algorithms as black-boxes that can be used to generate partial or full sequences, and survey algorithms that construct search spaces on the (partial-)sequence level or operate by drawing multiple samples.

## 4 Meta-generation algorithms

Some generation algorithms have the distinctive property of requiring access to a separate generation sub-routine. For instance, best-of-$N$ calls a generator to sample $N$ sequences from the language model. This sub-generator is interchangable; it can be freely chosen from top-$k$, temperature sampling, or any other sequence generator. We coin the term *meta-generation* to describe algorithms that call sub-generators, i.e.,

$$y \sim g(\cdot | x, \{g_1, \dots, g_G\}, \phi), \tag{23}$$

where $g$ defines the strategy used by the meta-generator, $\{g_1, \dots, g_G\}$ are sub-generators, and $\phi$ is a generic parameter for any other inputs (such as verifier or retrieval models) and hyperparameters (such as the number of tokens to generate). Since a meta-generator is itself a generation algorithm, i.e., a function that maps inputs to a distribution over outputs (§2.2), we will freely use $g(\cdot)$ for either a meta-generator or a token-level generator. We will often hide the parameters $\{g_1, \dots, g_G\}$ or make other parameters explicit based on the context. We identify four common strategies among meta-generators. In particular, we find that they can be classified into the categories of chained, parallel, step-level, and refinement-based meta-generators.

### 4.1 Chained meta-generators

The first meta-generation strategy chains multiple generators together. We start by explaining this idea in the context of prompted language models.

**Chaining prompted language models.** It is increasingly common to perform input-output tasks with a language model by specifying a prompt $z$,

$$y = f(x; p_\theta, z, \phi), \tag{24}$$

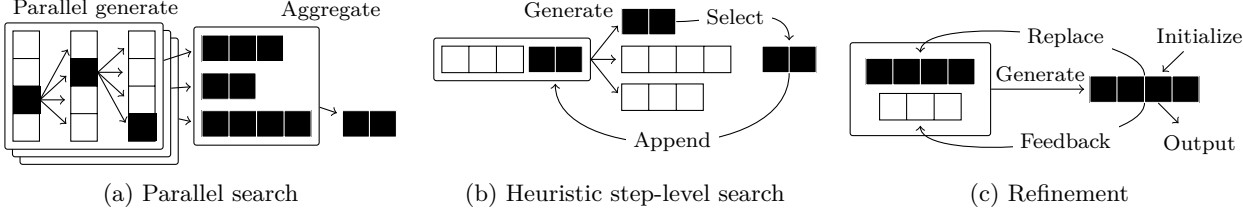

(a) Parallel search      (b) Heuristic step-level search      (c) Refinement

Figure 4: Three meta-generation patterns.

where $f(\cdot)$ is a deterministic generator, and the prompt $z$ is a sequence of tokens that specifies the desired behavior through a natural language instruction or input-output examples (Brown et al., 2020; Ouyang et al., 2022). For instance, given $z = $ `multiply the two numbers` and $x = 1432\ 293$, we can generate an output $y$ that contains an (attempted) solution. It is natural to compose the generator call with other operations, such as composing a generator that outputs Python code with a function that executes Python code.

Similarly, it is natural to combine multiple calls to generators, e.g., generating a story using:

$$y = f_3 \circ f_2 \circ f_1, \tag{25}$$

where $f_1$ generates a story outline, $f_2$ fills in the sections, and $f_3$ revises the story to meet a length constraint. Notice that the composition is itself a generation algorithm,

$$f(x; p_\theta, (f_1, f_2, f_3)), \tag{26}$$

i.e., a mapping from an input $x$, model $p_\theta$, and other parameters $\phi$, to an output (here, $\phi$ contains the generation algorithms $f_1, f_2, f_3$), or in general, a distribution over outputs $g(y|x, p_\theta, \phi)$. In general, we can view calls to generation algorithms as steps in a *program* whose execution yields a generated output. We can view the program $f(x; p_\theta, F)$, which calls generation algorithms $f' \in F$, as a *meta-generation algorithm*.

Related ideas appear in the literature under various names, including the programmatic view in Demonstrate-Search-Predict (DSP) and DSPy (Khattab et al., 2022; 2024), language model cascades (Dohan et al., 2022), LLM program (Schlag et al., 2023), and recently, scaffolding program (Zelikman et al., 2024b). We introduce the term meta-generation as an abstraction that is agnostic to the implementation of the underlying generator model(s) (which need not be LLMs), and to clarify the connection with other generation algorithms.

**Problem decomposition.** A variety of algorithms have adopted the chain pattern in order to decompose an input-output problem into multiple steps, with each step implemented by a language model or external function. As a motivating example, Chain-of-Thought (Wei et al., 2022) decomposes generation into generating a "chain-of-thought" followed by generating an answer i.e.,

$$z \sim g(z|x), y \sim g(y|x, z), \tag{27}$$

where $g$ is a generator, $x$ is an input, $z$ is an intermediate sequence (a "chain-of-thought"), and $y$ is an answer. It is instructive to view a variety of methods as generalizing this two-part decomposition to multiple intermediate sequences, $z_1, z_2, \ldots$, that involve calls to generators or external functions (Dohan et al., 2022). For instance, least-to-most prompting (Zhou et al., 2023a) first calls a generator to decompose a problem into sub-questions and then consecutively calls a generator to answer each sub-question, while Self-Ask (Press et al., 2023) additionally calls a search engine after generating each sub-question. Both of these are special cases of Demonstrate-Search-Predict (DSP) programs (Khattab et al., 2022). A wide range of methods can be seen as constructing alternative chained meta-generators, ranging from System 2 Attention (Weston & Sukhbaatar, 2023), which rewrites an input prior to generation to help the model refrain from attending to irrelevant information, to methods that decompose formal proof generation (Jiang et al., 2023). More generally, a number of tools such as LangChain (Chase, 2022) and MiniChain (Rush, 2023) provide domain-specific languages for declaring and executing chains involving prompted language models.

| Algorithm | Aggregation type | Scoring / transforming with |
|---|---|---|
| Best-of-N 24 | Rerank | LLM score or external score |
| Noisy-channel 163 | Rerank | Log-linear combination score |
| Majority voting 9 | Transform | Empirical vote frequency |
| Weighted majority voting 215 | Transform | Empirical distribution over answers |
| Self-consistency 220 | Transform | Marginal distribution over answers |
| Universal self-consistency 30 | Transform | Answer aggregation using an LLM generator |
| Branch-Solve-Merge 186 | Transform | Answer aggregation using an LLM generator / rule-based parsing |
| QE-fusion 217 | Transform | Answer contains spans from candidates |

Table 3: Parallel meta-generators.

## 4.2 Parallel meta-generators

Another pattern is to generate multiple trajectories in parallel, then merge the resulting terminal states to arrive at a final generated sequence. For instance, various *sequence-level generation algorithms* generate an *N-best list* $\{y^{(n)}\}_{n=1}^N \sim g$, then apply an *aggregation* function $h(y^{(1)}, \ldots, y^{(N)})$ to arrive at a final generated sequence. The $N$-best list of sequences might come from sampled generations, a beam search algorithm, or any other generator $y \sim g$ that generates full sequences. We discuss aggregation functions that rerank (§4.2.1) or transform (§4.2.2) the $N$-best list, then discuss sequence-level statistical rejection sampling (§4.2.3). Table 3 presents a brief summary of algorithms from the classes that we discuss.

### 4.2.1 Reranking algorithms

*Reranking* (or rescoring) is a classical approach (Collins, 2000; Huang & Chiang, 2007) originally developed for parsing and automatic speech recognition to achieve a trade-off between the computational complexity of MAP decoding and its tendency to rule out good hypotheses. A reranking algorithm orders an $N$-best list with a *reranking function* $h(y^{(1)}, \ldots, y^{(N)}) \to (y^{\sigma(1)}, \ldots, y^{\sigma(N)})$, then selects the top-$k$ ranked sequences. Reranking has recently found new applications in text generation (e.g., Cobbe et al. (2021); Stiennon et al. (2020); Krishna et al. (2022); Ni et al. (2023); Lightman et al. (2024)) by using various reranking functions and various sources of data to learn the reranking functions. A simple and effective method is *best-of-N*.

**Best-of-$N$.** Best-of-$N$ (Charniak & Johnson, 2005; Pauls & Klein, 2009) refers to generating an $N$-best list and picking the best sequence according to a scoring function.

**Definition 5** (Best-of-$N$: BoN$(x, g, v, N; \phi)$)**.** Let $g$ be a generation algorithm with output space $\mathcal{Y}$, and $v : \mathcal{Y} \to \mathbb{R}$ a scoring function. Assume that $\epsilon \in \phi$ governs the randomness in $g$. The best-of-$N$ generation algorithm is defined as:

$$f(x, g, v, N, \phi) = \underset{y^{(n)} | n \in \{1, \ldots, N\}}{\arg\max} \left\{ v(y^{(n)}) \mid y^{(n)} \sim g(\cdot | x), n \in \{1, 2, \ldots, N\} \right\}, \tag{28}$$

where each $y^{(n)}$ is a generated sequence.

Best-of-$N$ can be performed with any algorithm that can be used to generate a list of $N$ sequences, including temperature sampling, beam search, Viterbi decoding, or many others. In the context of language modeling, best-of-$N$ was developed for parsing (Charniak & Johnson, 2005; Pauls & Klein, 2009), and traditionally involved modifying a decoding algorithm originally developed to find the top-1 hypothesis so that it obtains the top-$N$ highest scoring decodings. An attractive property is that Best-of-$N$ usually incurs only a linear increase in computational complexity compared to top-1 decoding. In the context of LLMs, best-of-$N$ is amenable to black-box generators (e.g., accessed via an API), since it does not require knowledge of the generator for populating the $N$-best list. Modern instances of best-of-$N$ use learned scoring functions that are often themselves parameterized by LLMs. We discuss examples from reasoning and preference alignment.

**Best-of-$N$ in reasoning.** In some settings the goal is to generate correct sequences, such as a correct solution to a mathematical problem or a program that passes test cases. A common approach in these cases is to learn a *verifier* $v_\psi(x, y) \to [0, 1]$ that predicts the probability that an output $y$ is correct, and use it within Best-of-$N$. Doing so has seen success in mathematical reasoning (e.g., Cobbe et al. (2021); Uesato

et al. (2022); Lightman et al. (2024)), code generation (Ni et al., 2023), and other settings with similar properties. Naturally, the performance depends on the quality of the verifier, which we return to in (§5.1).

**Best-of-N in alignment.** Previously in §3.4, we discussed how the problem of aligning the distribution of generated text with a distribution of text preferred by humans can be framed as sampling from

$$q_*(y|x) = \frac{1}{Z(x)} p_\theta(y|x) \exp\left(\frac{1}{\beta} r(x,y)\right). \tag{29}$$

When a single high-reward sequence is desired (e.g., at low values of $\beta$), a natural strategy is to use best-of-$N$ with a learned approximation of the reward, $v_\psi(x,y)$, as the scoring function. In practice, this strategy is an effective alternative to reinforcement learning from human feedback (RLHF) methods (Gao et al., 2022; Beirami et al., 2024). For example, AlpacaFarm (Dubois et al., 2023) found that Best-of-1024 with a human-preference reward model was competitive with more standard decoding methods with a model trained using RLHF. A potential benefit is that Best-of-N does not require updating the model $p_\theta$'s parameters, at the expense of generation-time compute.

Best-of-N depends on the quality of the reward function, which is typically a learned function $v_\psi(x,y)$, termed a *reward model*. It can suffer from *reward over-optimization*–i.e., returning an undesired sequence that nevertheless receives high reward. Specifically, suppose that $q_*(y|x) \propto v_*(x,y)$, where $v_*$ perfectly captures the desired outcome of generation. Best-of-$N$ at high values of $N$ can be seen as approximating:

$$\underset{y\in\mathcal{Y}}{\arg\max}\, q_*(y|x) \approx \underset{y_n|n\in\{1,\ldots,N\}}{\arg\max}\, v_\psi(x,y_n), \tag{30}$$

where $y_n \sim g$. In practice, the learned model $v_\psi$ typically does not match $v_*$, especially on out-of-distribution sequences, so best-of-$N$ may find sequences that "overoptimize" the reward (Gao et al., 2022).

**Noisy-channel reranking in Neural Machine Translation.** A wide range of reranking methods precede the era of large language models. A classic approach is a noisy-channel model (Brown et al., 1993). *Noisy-channel* means that the observed output from the system (e.g., a machine translation system) is distorted by some unknown noise pattern (i.e., noisy channel). If we consider $p_\theta(y|x)$ as the probability of the translation $y$ of the source language text $x$, then Bayes rule suggests the following relationship: $p_\theta(y|x) \propto p(x|y)p(y)$, where $p(x|y)$ is a channel model, and $p(y)$ is the target language LM.

As an example from the literature, Och & Ney (2002); Ng et al. (2019) propose to use the following log linear combination to rerank translation candidates:

$$s_{\text{noisy-channel}}(y) = \log p(y|x) + \lambda_1 \log p(x|y) + \lambda_2 \log p(y), \tag{31}$$

where the log-linear coefficients $\lambda_1$ and $\lambda_2$ are tuned empirically on a development set. Therefore, the reranking function $h$ in this case is defined so that the order of candidates is given by a decreasing order of noisy channel scores $s_{\text{noisy-channel}}$ computed for every translation candidate.

### 4.2.2 Transformation algorithms

In contrast to reranking elements of the $N$-best list, other algorithms transform the list into a new sequence which might not be part of the $N$-best list itself. For instance, mathematical question answering is an example of a task where the potential outputs (answers to math questions) are produced as *part* of much longer decoded sequences from the LLM. In other cases we might draft $N$ summaries, then synthesize them into a new, final summary. This requires a transformation of the $N$ summaries rather than a simple reranking.

**Majority voting.** Majority voting (or self-consistency (Wang et al., 2023b)) processes a $N$-best list and counts how each of the candidates $y^{(i)}$ votes towards a different set of outputs $(a_1, \ldots, a_K)$:

$$h(y^{(1)}, \ldots, y^{(N)}) \rightarrow (c(y^{(1)}), \ldots, c(y^{(N)})), \tag{32}$$

where $c : \mathcal{Y} \to 1, \ldots, K$ is a voting function that maps from sequence space to an output from $(a_1, \ldots, a_K)$. Second, it selects the output that received the largest number of votes:

$$\hat{a} = \arg\max_k \sum_{j=1}^{K} \sum_{i=1}^{N} \mathbb{I}(c(y^{(i)}) = j). \tag{33}$$

**Weighted majority voting.** Often it is beneficial to incorporate a reward model $v_\psi$ into voting. The reward model can bias the distribution or break ties. The final output selection is done by aggregating (e.g., summing) scores associated with the votes:

$$\hat{a} = \arg\max_k \sum_{j=1}^{K} \sum_{i=1}^{N} v_\psi(y^{(i)}) \mathbb{I}(c(y^{(i)}) = j). \tag{34}$$

Let $y = (z, a)$ denote an output decomposed into a sequence $z$ (e.g., a reasoning chain) and an answer $a$. Wu et al. (2024) prove that as $N \to \infty$, weighted voting accuracy on a dataset of $M$ examples converges to,

$$\frac{1}{M} \sum_{i=1}^{M} \mathbb{I}\left[a_i^* = \arg\max_a \sum_z v_\psi(z, a) g(a, z|x)\right]. \tag{35}$$

This is the accuracy obtained after marginalizing out the sequences $z$. Voting can be seen as sample-based approximation based on $N$ generated intermediate sequences (Wang et al., 2023b). Moreover, weighted voting will outperform voting when $v \cdot g$ has more total mass on correct answers than $g$ (Wu et al., 2024).

Figure 6 shows a performance comparison of best-of-$N$, majority voting, and weighted majority voting on a standard mathematical reasoning benchmark.

**Minimum Bayes Risk decoding.** Rather than seeking the most probable sequence as in MAP decoding, Minimum Bayes Risk algorithms aim to find the best sequence in terms of a pairwise *utility* function $u(y, y')$:

**Example 2** (Minimum Bayes Risk algorithm)**.** A Minimum Bayes Risk (MBR) decoding algorithm refers to an algorithm of the form (Bickel & Doksum, 1977; Kumar & Byrne, 2004):

$$f(x) \triangleq \arg\max_{y' \in \mathcal{Y}} \sum_{y \in \mathcal{Y}} u(y, y') p_*(y|x), \tag{36}$$

where $u : \mathcal{Y} \times \mathcal{Y} \to \mathbb{R}$. A dependence on $p_\theta$ is introduced in specific instances of MBR decoding algorithms.

The MBR objective is motivated by decision theory. Intuitively, it can be thought of as seeking an output with highest average "similarity," as measured by the utility function $u$, to other candidates, particularly those assigned high probability under $p_*$.

Various algorithms provide approximate solutions to the Minimum Bayes Risk (MBR) objective. They typically consist of providing a utility $u(\cdot, \cdot) \to \mathbb{R}$, populating a *hypothesis set* $\mathcal{Y}_h$ using a generator, and populating a *evidence set* $\mathcal{Y}_e$ to estimate the risk of each hypothesis:

$$\hat{y} = \arg\max_{y' \in \mathcal{Y}_h} \frac{1}{\mathcal{Y}_h} \sum_{y \in \mathcal{Y}_e} u(y, y'). \tag{37}$$

The hypothesis set is typically akin to an N-best set, populated by calling a generator $\{y^{(n)}\}_{n=1}^{N} \sim g(\cdot|x)$. Simple strategies sample from the model, $y^{(n)} \sim p_\theta$. Others take the best-$k$ outputs from a ranked list of generations, or use more sophisticated strategies such as iteratively adding hypotheses or transforming them (González-Rubio et al., 2011; González-Rubio & Casacuberta, 2013). Freitag et al. (2023) investigate the impact of the underlying sampling strategy, finding variation across strategies, with epsilon sampling performing best for machine translation. The evidence set is typically sampled from a generator, or set to the hypothesis set to save on computation. Finally, the metric impacts performance. For example, MBR with a metric tends to inflate performance on that metric, sometimes by gaming it (Freitag et al., 2023).

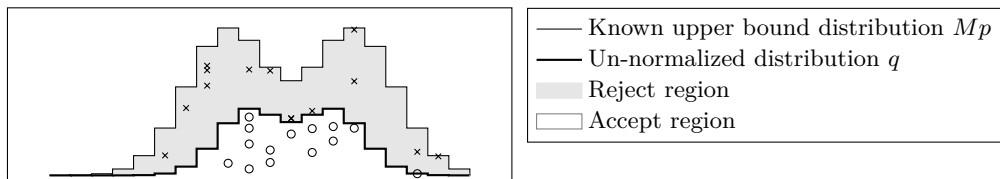

Figure 5: In rejection sampling, the aim is to sample from a distribution $q$ whose normalizing constant is unknown. To do so, use a known distribution $p$ that serves as an upper bound for the unknown distribution when scaled by a constant, i.e., for some constant $M$ and all values $y$, $Mp(y) \geq q(y)$. Next, obtain a sample $y \sim p$ and accept this sample with probability $q(y)/Mp(y))$, otherwise reject the sample and repeat the process. This is equivalent to sampling from $q$.

MBR methods have a rich history in the machine translation and speech recognition literature (Goel et al., 2004; Heigold et al., 2005; GOEL, 2003; Kingsbury et al., 2012; Eikema & Aziz, 2020), and have also been applied across other tasks (Shi et al., 2022; Suzgun et al., 2023). Interestingly, Bertsch et al. (2023) show that self-consistency and other voting techniques are special cases of MBR. For example, weighted voting (and as a special case, voting) corresponds to a utility that checks if two answers match,

$$u(y, y') = \mathbb{I}\left[a = a'\right] \cdot v(y'),$$ (38)

where $y = (z, a), y' = (z', a')$, and $v$ is the weighted voting scoring function. In general, there are several other dimensions along which MBR methods are categorized. We refer the reader to Bertsch et al. (2023) for a further in-depth study and a taxonomy of MBR methods.

**Generate-and-transform.** In general, we can view the algorithms above as first generating an $N$ best list, followed by transforming the $N$ best list using a transformation $h(y^{(1)}, \ldots, y^{(N)})$, such as voting or one that internally estimates risk. Rather than hand-designing the transformation, recent methods explore using language models themselves. For instance, universal self-consistency (Chen et al., 2023c) prompts a language model to generate a final sequence given the $N$-best list, which can avoid the aforementioned issue of parsing sequences into an answer. Branch-solve-merge (Saha et al., 2023) transforms an input into $N$ different prompts, generates with those prompts, then merges the results by prompting a language model.

### 4.2.3 Sequence-level rejection sampling

Previously we discussed the goal of designing a generation algorithm that samples from a target distribution $q_*$ (§2.2.3). A related pattern is using a stochastic sequence generator $y \sim g$ to sample from $q_*$ using rejection sampling. This involves sampling multiple sequences from $g$ and is thus akin to a parallel meta-generator.

Specifically, statistical rejection sampling is a technique for sampling from a target distribution $q_*$ with an unknown normalizing constant. This is accomplished by first sampling from a known distribution $y \sim g$ which serves as an upper bound for $q_*$, (e.g., for some constant $M$, $Mg(y) \geq q_*(y)$), then accepting the sample with probability $q_*(y)/Mg(y)$. Figure 5 illustrates this process. Rejection sampling is a useful tool for sampling from a specified target distribution over an intractably large support, e.g., the set of sequences.

One example of *sequence-level* rejection sampling for LMs is sampling valid JSON strings from an LM. The space of valid JSON strings is infinite and the normalizing factor is unknown, but we can sample from this distibution by first sampling from the LM distribution $p_\theta$, then rejecting any string that is not valid JSON. Here, the *un-normalized* distribution we are sampling from is

$$q_*(y) \propto \begin{cases} p_\theta(y) & y \text{ is valid JSON} \\ 0 & \text{Otherwise} \end{cases},$$

and we must use rejection sampling since the normalization term is unknown.

**Best-of-$N$ and rejection sampling.** Above we introduced best-of-$N$ as a deterministic algorithm (Definition 5). Another view is that calling best-of-$N$ with a stochastic generator $g$ is itself a stochastic generator,

$$y \sim \textsc{Bon}(p_\theta, g, N, v), \tag{39}$$

where $\textsc{Bon}$ means generating $N$ sequences $y^{(1)}, \ldots, y^{(N)} \sim g$, then selecting the sequence with the highest score $v$. This idea has been termed the *best-of-$N$ policy* (Stiennon et al., 2020; Gao et al., 2022). Interestingly, Gao et al. (2022) find that the best-of-$N$ policy may give similar reward maximization to reinforcement learning, though with a different pattern of divergence from the underlying language model. Their analysis uses an analytical expression for the KL divergence from (Stiennon et al., 2020), $\mathrm{D}_{KL}\left(\pi_{\textsc{Bon}} \| p_\theta\right) \approx \log N - (N-1)/N$. Beirami et al. (2024) show that this expression is an upper bound on the actual KL divergence and propose an estimator that empirically provides a tighter approximation.

Finally, $y \sim \textsc{Bon}$ can be understood as internally performing rejection sampling (Stiennon et al., 2020). We refer the reader to Liu et al. (2024c) for a more detailed discussion of this connection, as well as an improved algorithm that builds on the connection between rejection sampling and best-of-N.

**Pseudo-rejection sampling.** Several decoding methods employ various forms of *pseudo*-rejection sampling. One example of this is Li et al. (2024a), where the authors sample a set of $k$ outputs from the LM, compute the "value" of each of these outputs, and then sample from the output set by interpreting the values as logits. As $k$ tends toward infinity, this method approaches sampling from the value function with a regularization term that keeps the distribution close to the LM distribution. Another method based on a similar construct is the one of Zhao et al. (2024b). Such methods can also be interpreted as sampling importance resampling, in the spirit of sequential Monte-Carlo sampling algorithms (Douc et al., 2014). Pseudo-rejection sampling is often employed when the prerequisites for rejection sampling are not met, for instance when there is no known upper bound on the target distribution.

### 4.3 Step-level search algorithms

Next, we discuss meta-generation algorithms that implement classical search algorithms by calling generators. To introduce these, it is helpful to view generation as navigating a state space $s \in \mathcal{S}$ by taking actions $a \in \mathcal{A}$ using a generator, and receiving new states from an environment $\mathcal{E} : \mathcal{S} \times \mathcal{A} \to \mathcal{P}(\mathcal{S})$, yielding a trajectory $(s_0, a_1, s_1, \ldots, a_T, s_T)$. The start state $s_0$ contains the input to the generation algorithm, i.e. $x \in s_0$, while the terminal state contains the output of the generation algorithm, i.e. $y \in s_T$. Generation consists of running the resulting process until reaching a terminal state.

As a basic example, recall that greedy decoding is defined as:

$$\hat{y}_t = \arg\max_{y_t \in \mathcal{V}} p_\theta(y_t | \hat{y}_{<t}, x), \tag{40}$$

for $t = 1, \ldots, T$. The search perspective interprets this as taking next-token actions $\hat{y}_t$ given states $(x, \hat{y}_{<t})$, a generator that selects the most probable next-token from $p_\theta$, and an environment that appends a next-token to form a state $(x, \hat{y}_{<t} \circ \hat{y}_t)$. Since greedy decoding is an approximate MAP decoding algorithm, it aims to end in a state that maximizes $p_\theta(y|x)$. In other cases the environment is less trivial, such as those involving code execution and visual observations (Shinn et al., 2023; Zhou et al., 2023b). Many algorithms can be recovered by varying the states, actions, environment, and/or generator.

In particular, reasoning tasks such as mathematical problem solving or theorem proving have served as a testbed for developing step-level search algorithms. In these tasks, the final output (a solution or a proof) naturally decomposes into 'steps', $y = (\mathbf{y}_1, \ldots, \mathbf{y}_T)$, where each $\mathbf{y}_t$ is itself a sequence of tokens. One can then consider a partial solution $\mathbf{y}_{<t}$ as the state $s_t$, and generating a next-step $\mathbf{y}_t$ as the action. The environment appends the next-step to the partial solution, $\mathbf{y}_{<t} \circ \mathbf{y}_t$. There is also a natural notion of success (i.e., a correct answer, a valid proof), leading to the idea of a *value function* $v(s_t) \to [0, 1]$ that is used to predict whether a solution-so-far will eventually be correct (or in general, predict the expected reward of the state).

Several algorithms maintain a queue of states that contain partially generated outputs, and iteratively select states for exploration. Exploring a state involves expanding the state's partial output and scoring the

| Method | Search | State | Generation | Value $v(s_t)$ | Tasks |
|---|---|---|---|---|---|
| gpt-f Proof Search [176] | Best-first | Proof-so-far | Proof step | $\log p_\theta$ | Formal proving |
| gpt-f +outcome [176] | Best-first | Proof-so-far | Proof step | $v_\psi \approx \mathbb{E}(\texttt{success})$ | Formal proving |
| Proofsize Search [177] | Best-first | Proof-so-far | Proof step | $v_\psi \approx \mathbb{E}(\texttt{length})$ | Formal proving |
| Stepwise++ [224] | Beam | Proof-so-far | Proof step | $\log p_\theta + n\text{-grams}$ | Informal proving |
| Self-Evaluation [233] | Beam | Steps-so-far | Reasoning step | $\log p_\theta + \text{LLM}$ | Multi-step correctness |
| Reward Balanced Search [229] | BFS-like | Steps-so-far | Reasoning step | $v_\psi \approx \mathbb{E}(\texttt{correct})$ | Multi-step correctness |
| Tree-of-Thought [240] | BFS/DFS | Steps-so-far | Generation step | Prompted LLM | Multi-step generation |
| Graph-of-Thought [14] | BFS/DFS | Steps-so-far | Generation step | Prompted LLM | Multi-step generation |
| HyperTree Proof Search [119] | MCTS | Proof-so-far | Proof step | $v_\psi \approx \mathbb{E}(\texttt{success})$ | Formal proving |
| AlphaLLM [211] | MCTS | Steps-so-far | Reasoning steps | $v_\psi \approx \mathbb{E}(\texttt{correct})$ | Multi-step correctness |
| Reasoning via Planning [78] | MCTS | Steps-so-far | Generation step | Prompted LLM | Multi-step generation |

Table 4: Survey of step-level search methods.

expanded output with a value function $v(s_t)$. The scores are then used to prune or prioritize states for the next iteration. Conceptually, step-level search is typically a tree search, consisting of states as nodes and actions plus environment transitions as edges. Although the algorithms below typically contain domain-agnostic ideas, we will ground the discussion below by discussing reasoning tasks as the running examples.

**Warmup: token-level beam search.** Traditional beam search (Graves, 2012; Sutskever et al., 2014) maintains a queue of prefixes $\{y_{<t}^k\}_{k=1}^K$ termed a *beam*, expands each prefix using each possible next-token, $\{y_{<t}^k \circ y_t \mid k \in \{1, \ldots, K\}, y_t \in \mathcal{V}\}$, scores each expanded prefix using $\log p_\theta(y_{<t}^k \circ y_t | x)$, and prunes the queue by keeping only the top-$K$ scored expansions for the next iteration. In this case, the value function is $\hat{v}(y_{<t} \circ y_t) = \log p_\theta(y_{<t} \circ y_t | x)$, and it is used to prune states by selecting the top-K expanded prefixes. Traditional beam search operates on the token-level, using the specific strategy of expanding each possible next-token, which assumes access to primitive operations (e.g., next-token distributions).

**Partial sequence expansion.** We can consider higher-level algorithms that operate on the partial sequence (i.e., "step") level rather than the token level, and call an arbitrary generator to expand states, e.g., $\{\mathbf{y}_t^{(k)}\}_{k=1}^K \sim g(\cdot | s_t)$. For example, stepwise beam search (Welleck et al., 2022; Xie et al., 2023) performs a beam search over steps of mathematical problems or proofs. Tree-of-thoughts (Yao et al., 2023) considers a beam search over generated steps that include additional "thought" sequences. Potential benefits of partial sequence expansion over traditional beam search include efficiency due to executing the value function less often, and not requiring access to all of a model's next-token probabilities.

**Alternate search strategy.** Another axis of variation is the underlying search strategy. Beam search is a pruned breadth-first search, which has been used with contemporary LLMs in methods such as stepwise beam search (Welleck et al., 2022), tree-of-thoughts (Yao et al., 2023), and recently in Snell et al. (2024). However, other search algorithms are available, such as best-first search, used in the context of formal theorem proving (Polu & Sutskever, 2020; Polu et al., 2023; Yang et al., 2023a). Formal theorem proving has a natural decomposition of outputs (i.e., a proof) into steps (termed "tactics"), which has historically made it a fruitful testbed for more advanced search algorithms. For example, HyperTree Proof Search (Lample et al., 2022) draws on Monte-Carlo Tree Search (MCTS)—a method that was central to the impactful Go playing system AlphaGo (Silver et al., 2016; 2017)—which prioritizes states according to a confidence bound and scores states by rolling out trajectories. Recently, similar ideas have been adapted to other LLM generation tasks. For instance, Reasoning via Planning (Hao et al., 2023) and ThoughtSculpt (Chi et al., 2024) incorporate MCTS selection and rollouts for several tasks. However, there are a few key differences between environments in which MCTS has excelled (such as Go) and general-purpose language generation. For example, MCTS explores based on visit counts in the search tree, which can require a large number of simulations (for instance, AlphaGo Zero used 1,600 simulations per move (Silver et al., 2017)). This can be prohibitively expensive with large language models. Second, Go has a natural decomposition into steps, and a reliable terminal reward (i.e., win vs. lose) that can aid in training a value function.

**Alternate value functions.** Another axis of variation is the choice of value function. For instance, in traditional beam search, a value function can be manually designed to score candidates $(y_{<t} \circ y_t)$ more highly

if they contain desired $n$-grams (Lu et al., 2022), while others use learned heuristics $v_\psi(y_{\leq t})$ trained to predict a property of the full sequence (e.g., its style score or correctness) (Yang & Klein, 2021). In formal theorem proving, it is common to train a value function that predicts whether a proof from the current state will eventually succeed (Polu & Sutskever, 2020; Lample et al., 2022) or a related statistic such as the expected proof length (Polu et al., 2023). In the context of large language models, a recent trend is to use a prompted language model to evaluate states (Yao et al., 2023). As mentioned previously, to achieve better results one can train a model that predicts whether the solution-so-far will eventually be correct, $v_\psi(y_{<t}) \rightarrow [0, 1]$ (or more generally, predict the expected reward from the current state), termed a process-based verifier (Uesato et al., 2022). Recently, Reward Balanced Search (REBASE) (Wu et al., 2024) allocates its exploration budget to nodes based on their process-based verifier scores. Liu et al. (2024b) repurposes the value function from proximal policy optimization (Schulman et al., 2017) as a MCTS value function. Refer to Hao et al. (2024) for additional discussion, including a library implementing several of the algorithms above.

### 4.4 Refinement algorithms

A *refinement* algorithm consists of (1) an initial generator $g_0$, (2) an information source $h$, (3) a refiner $g$:

$$y^{(0)} \sim g_0(y|x), \tag{41}$$

$$z^{(t)} \sim h(z|x, y^{(<t)}, z^{(<t)}), \tag{42}$$

$$y^{(t)} \sim g(y|x, y^{(<t)}, z^{(\leq t)}). \tag{43}$$

Intuitively, the refiner generates a "revised" output $y^{(t)}$ given previous versions $y^{(<t)}$ and extra information $z^{(\leq t)}$, such as feedback or environment observations. The algorithm alternates between receiving information, $z \sim h$, and refining, $y \sim g$, until a stopping condition is met. Refinement algorithms vary based on choice of initial generator, the refiner, the content and source of extra information $z$, and the stopping condition.

**Learned refiners.** Self-correction, introduced by Welleck et al. (2023), provides a recipe for training a refiner model $p_\theta(y^{(t)}|x, y^{(t-1)}, z^{(t)})$ which iteratively refines an output to improve the score from a reward function $r(x, y)$ using $(z^{(t)}, y^{(t-1)}, y^{(t)})$ examples collected from model trajectories. Here $z$ is either $0/1$ (apply the refiner or do not apply the refiner) or a feedback string. $z$ is assumed to be given to the system at generation time, which is a limitation for some tasks (e.g., we often do not know whether a mathematical solution should be revised). GLoRe (Havrilla et al., 2024) relaxes this limitation by training a verifier to determine whether to apply the refiner, and to localize per-step errors.

**Prompted refiners.** A second option is to parameterize the refiner using a prompted language model,

$$y^{(t)} \sim g_\theta\left(y|P_{\text{refine}}(x, y^{(<t)}, z^{(\leq t)})\right), \tag{44}$$

where $g_\theta$ is a generation algorithm that involves prompting a model $p_\theta$ with a prompt $P_{\text{refine}}(x, y^{(<t)}, z^{(\leq t)})$, as introduced in Self-Refine (Madaan et al., 2023) and Reflexion (Shinn et al., 2023). This allows the initial generator and the refiner to share a single language model that is not necessarily tuned for a specific task.

**Prompted feedback.** It is common for the information $z \sim h$ to include "feedback" on a preceding version $y$, with the feedback being a sequence of tokens generated with a prompted language model:

$$z^{(t)} \sim h_\theta\left(z|P_{\text{feedback}}(x, y^{(<t)}, z^{(<t)})\right). \tag{45}$$

This feedback is often also termed "critique" (Matiana et al., 2021; Castricato et al., 2022; Bai et al., 2022; Saunders et al., 2022). Self-Refine (Madaan et al., 2023) shares $\theta$ across the feedback provider, refiner, and initial generator, yielding a refinement algorithm given only a model $p_\theta$ and 3 prompts. Similarly, Reflexion (Shinn et al., 2023) uses a prompted feedback provider. In these cases, the feedback is termed *self-feedback* or *self-reflection*.

**Environment feedback.** As we discussed previously, the search perspective treats generation as a trajectory $(s_0, a_1, s_1, \ldots, a_T, s_T)$, with state transitions determined by an environment $\mathcal{E} : \mathcal{S} \times \mathcal{A} \to \mathcal{P}(\mathcal{S})$. In some cases the environment transitions are nontrivial, such as executing generated code or clicking a link in a webpage. We can view the resulting observations (e.g., code execution results or an image of a webpage) as extra information $\tilde{z}^{(t)} \in s^{(t)}$ that is contained in the state. This information can be passed to the feedback provider to generate feedback $\bar{z}$, and the refiner (e.g., via prompts):

$$\bar{z}^{(t)} \sim h_\theta \left( \bar{z} | P_{\text{feedback}}(x, y^{(<t)}, \tilde{z}^{(\le t)}, \bar{z}^{(<t)}) \right), \tag{46}$$

$$y^{(t)} \sim g_\theta \left( y | P_{\text{refine}}(x, y^{(<t)}, \tilde{z}^{(\le t)}, \bar{z}^{(\le t)}) \right), \tag{47}$$

meaning that each iteration refines based on new environment information (e.g., code execution results).

Reflexion (Shinn et al., 2023) adopts this perspective of generation as a trajectory involving an environment, with the refiner akin to an "actor". The idea has since been adapted to digital agents (Kim et al., 2023; Pan et al., 2024), code (Chen et al., 2024b; Shi et al., 2024b), and other environments (Pan et al., 2023).

**Does refinement work?** Notice that a refinement algorithm is a 3-tuple $(g_0, h, g)$. Intuitively, if the information source $h$ adds new information beyond that contained in the initial generator $g_0(\cdot | p_\theta)$ to the refinement algorithm, it is plausible that a refinement algorithm can outperform the initial generator alone. Hence, in our discussion we distinguish between algorithms that receive information external to the generator at inference time, and those that do not receive external information at inference time (termed *extrinsic* and *intrinsic* refinement or self-correction, respectively (Huang et al., 2024; Kumar et al., 2024)).

For extrinsic refinement, it is plausible that there are information sources which add new information beyond that in $g_0(\cdot | p_\theta)$, and hence lead to a potential gain with refinement. For instance, if $z \sim h$ contains a compiler error, test case results, or an image of webpage after it is clicked, the refiner plausibly receives new information. Similarly, if $z \sim h$ represents feedback, and the feedback comes from a source outside of the model $p_\theta$ (e.g., a human, a model with additional parameters, supervision, or a different objective), the feedback function may be expected to add new information beyond that in the initial generator. Indeed, refinement has seen success in code generation, where a code interpreter or compiler provides feedback (e.g., Chen et al. (2023d)), or when a retriever or larger model (Olausson et al., 2024) provide feedback. In a similar vein, Reflexion (Shinn et al., 2023) finds that without code execution, refinement yields little to no gain on code generation. What factors contribute to the potential improvements, aside from the general notion of "adding new information"? Taking a compiler as an example, the compiler can potentially (i) evaluate correctness, (ii) localize errors, and (iii) provide other semantic information related to an error (e.g., an error type). A parallel method such as Best-of-$N$ (see §4.2) can only use this information to prefer one sequence over another. A refiner, in contrast, can implement an arbitrary transformation that acts upon the information. That said, fully understanding the factors that enable refinement remains an open question.

For intrinsic refinement, we first consider refinement algorithms that prompt a single model for initial generation, feedback, and refinement (e.g., Madaan et al. (2023)). In these cases, the efficacy of refinement has been mixed. For example, Huang et al. (2024) and Tyen et al. (2024) find that such methods often fail to improve on reasoning tasks. The failure likely stems from the difficulty of a model evaluating the correctness of its own outputs (Huang et al., 2024). Similarly, Havrilla et al. (2024) observe that even learned verifiers trained to predict correctness can have high false positive rates that trigger spurious refinements. In these cases, we can view the information source as being too noisy for the refiner to reliably act upon.

Finally, we consider intrinsic algorithms that aim to train a refiner. The key difference with prompted methods is that the refiner can receive new information through the training procedure. For example, code correctness might be used as a reward signal to train a refiner that maps buggy code to correct (high reward) code. Since the refiner is typically a generic sequence-to-sequence model, the key question is whether such a refiner can be trained in practice. Self-corrective learning (Welleck et al., 2023) framed training as generating $(x, y, y')$ examples online with the initial generator, refiner, and a reward function, updating the refiner by fine-tuning on the examples, then repeating the process. However, recently Kumar et al. (2024) identified a *behavior collapse* phenomenon, in which the refiner learns to ignore the intermediate output $y$. This

motivated a reinforcement learning procedure that alleviates behavior collapse and leads to some degree of intrinsic self-correction. More generally, learning intrinsic self-correction remains an active research direction.

In summary, refinement presently works best for tasks that either have rich environment feedback or can be reliably evaluated by current language models. As language models improve as verifiers, the range of tasks for which refinement is effective will likely grow. However, this may be paired with improvements in the abilities of the initial generator $g_0$, potentially to the extent that intrinsic refinement is no longer necessary.

## 5 Incorporating external information

Next, we consider what kinds of information a generation or meta-generation algorithm incorporates outside of the language model, such as other models or tools. Algorithms use external information by calling operations beyond primitive operations from $p_\theta$ (e.g., those from another model), or through assumptions on the inputs or outputs. We comment on common patterns related to incorporating external information.

### 5.1 Multiple models

A variety of generation algorithms incorporate multiple models. More formally, recall that in (§2.2) we defined a generation algorithm as a function that maps an input $x$, model $p_\theta$, and other inputs $\phi$ to a distribution $g(y|x; p_\theta, \phi)$. A generator *uses multiple models* if $\phi$ contains other models (e.g., an additional language model), and operations from the model are used in the algorithm. In this sense, the external models can add new information beyond that contained in $p_\theta$ to the generator.

**Small and large language models.** A notable pattern is using a small language model to either adjust a model's distribution or to speed up generation. Lu et al. (2023) train a small model $p_\beta$ with reinforcement learning such that it adjusts the next-token probabilities of a larger model $p_\theta$ to maximize a reward function. The models are combined into a token-level "product of experts" (Liu et al., 2021a),

$$p'(y_t|y_{<t}, x) = \frac{1}{Z} p_\theta(y_t|y_{<t}, x) p_\beta(y_t|y_{<t}, x)^\alpha, \tag{48}$$

where $p_\beta$ is a separate language model, $\alpha \in \mathbb{R}$, and $Z \in \mathbb{R}$ is a normalization constant. Liu et al. (2024a) adopt a similar idea but with supervised finetuning of $p_\beta$. In order to amplify the improvement of a large, strong model over a small, weak one, contrastive decoding (Li et al., 2023a) defines a scoring function for beam search that returns the difference between the likelihood under the model $p_\theta$ with that of a smaller language model $p'_\theta$,

$$s(y_{<t} \circ y_t) = \log p_\theta(y_t|y_{<t}) - \log p'_\theta(y_t|y_{<t}), \tag{49}$$

along with a truncation criterion that sets the score to zero for some tokens. Intuitively, the smaller model often has larger model errors on unfavorable tokens (e.g., assigning more probability to tokens leading to repetition or incoherence compared to $p_\theta$). Assuming there is a nontrivial difference in probability assigned to these tokens, the score will reduce their prevalence in generated texts.

Finally, *speculative decoding* (Leviathan et al., 2022) is motivated by speeding up generation, which we will discuss further in (§7). It uses a small *draft* model to propose generations that are verified or rejected in parallel by the larger model $p_\theta$, hence speeding up generation when the rejection rate is not too high.

**Scalar feedback models.** A common pattern is learning a *verifier model* $v_\psi(x, y) \to [0, 1]$ that predicts the probability that a generation is correct (Cobbe et al., 2021). The verifier can be used to select outputs in best-of-$N$, or for weighted majority voting (Li et al., 2023b). This pattern is particularly suitable for mathematical problem solving and code generation (Ni et al., 2023), which have well-defined notions of correctness. Several works have iterated on the verifier model's design and learning procedure. Uesato et al. (2022) show that a verifier trained to predict the correctness of each *step* in an output (termed a *process-based* verifier) can outperform a verifier trained to predict the correctness of a full solution (termed an *outcome-based* verifier), and Lightman et al. (2024) obtain new human annotations for a process-based verifier. Math Shepherd (Wang et al., 2023a) propose a method for obtaining supervision from generations.

An underlying idea is that a generation algorithm that incorporates the verifier may have capabilities beyond those of $p_\theta$. This may be due to additional supervision, or factors that stem from the intuitive idea that *evaluation is often easier than generation.* For example, Sun et al. (2024a) show that weighted majority voting with a verifier can improve the generator's ability to generalize to harder problems.

More generally, learning a verifier is a special case of learning a scalar reward model $v_\psi(x, y)$ that can be used to select or score outputs. For instance, in (§4.2.3) we discussed using a reward model of human preference ratings to select outputs in best-of-$N$ (Stiennon et al., 2020; Ouyang et al., 2022; Touvron et al., 2023). As we discussed previously (§4.2.3), we can view this as shifting the generation distribution.

**Information conveyed in prompts.** Rather than using a separate model in a multi-model system, it is now common to parameterize different models by providing different prompts. For instance, we can obtain a feedback model by prompting a model to provide feedback. It is important to note that the prompts can add new information to the generation algorithm. As another use of multiple prompts, Kim et al. (2024b) use the logits obtained with a perturbed input prompt to adjust the logits used for token-level generation.

## 5.2 External environment information

More generally, generation algorithms can incorporate information from an external environment.

**Calling an external tool.** Certain functionality such as reliably performing a calculation or a web search may either be outside of a model's capabilities or inefficient to perform with the language model. A natural alternative is to issue a call to an external routine that performs the functionality at generation time.

One way to do this is through special tokens that denote a call to the routine, followed by replacing the prefix with the result. For instance, suppose the preceding tokens $y_{<t}$ include `[CALC]4+4[/CALC]`. Then at step $t$ of a token-level decoding algorithm, a calculator would be called on the query `4+4`, and in subsequent steps, the prefix $y_{\leq t}$ would contain the result `8`, along with possible reformatting (e.g., removing `[CALC]`).

A second common use of an external routine is as a verifier following the generation of a full sequence. For instance, in language-model based theorem proving the proof assistant is used to verify or reject generated proofs, while in code generation it is common to execute test cases. More generally, the notion of "tool use" (i.e., calling external programs) is now widespread, and has been incorporated into libraries such as LangChain (Chase, 2022) and products. Refer to Wang et al. (2024b) for further discussion.

**Receiving observations from an environment.** The search perspective framed generation as a sequential decision making process that involves observations from an environment (§4.3). A notable application area is code generation, which has natural environment information (e.g., interpreters, compilers). For instance, Lever (Ni et al., 2023) feeds execution results into a reward model used for best-of-N, while Self-Debugging (Chen et al., 2024b) incorporates error messages into refinement. A recent line of work tailors generation algorithms to language-conditioned digital agents–models that operate on diverse observation spaces $\mathcal{X}$ such as images of web pages, and output sequences $y$ representing actions–including variants of refinement (Shinn et al., 2023) combined with learned evaluators (Pan et al., 2024).

## 6 Token cost and performance analysis

A natural question is the cost of executing a given meta-generator, and its relationship with performance. There are several ways to measure cost, including the number of tokens generated, the overall compute used during generation, or the runtime. In some cases, we would like to design an algorithm that improves as we add more cost, such as improving problem solving ability by generating more tokens. In other cases, we would like to minimize the cost at a fixed level of performance.

| Method | Input | Output | External | Cost Params |
|---|---|---|---|---|
| Ancestral Sampling | $T_{\text{in}}$ | $T$ | – | – |
| Reranking (general) | $T_{\text{in}} * N$ | $T * N$ | $N * C_s$ | $N, C_s$ |
| Best-of-$N$ (log-p) | $T_{\text{in}} * N$ | $T * N$ | – | $N$ |
| Best-of-$N$ (LLM sequence scorer) | $T_{\text{in}} * N$ | $T * N$ | $N * (T_{\text{in}} + T + 1)$ | $N$ |
| Transformation (general) | $T_{\text{in}} * N$ | $T * N$ | $C_t$ | $N, C_t$ |
| Self-consistency | $T_{\text{in}} * N$ | $T * N$ | – | $N$ |
| Weighted SC (seq. scorer) | $T_{\text{in}} * N$ | $T * N$ | $N * C_s$ | $N, C_s$ |
| Step-level beam (log-p) [224] | $T_{\text{in}} * N_b * N_e * S$ | $T_s * N_b * N_e * S$ | – | $N_b, N_e, S$ |
| Step-level beam (seq. scorer) [240] | $T_{\text{in}} * N_b * N_e * S$ | $T_s * N_b * N_e * S$ | $N_b * N_e * S * C_s$ | $N_b, N_e, S, C_s$ |
| Step-level DFS (seq. scorer) [240] | $T_{\text{in}} * N_e * S$ | $T_s * N_e * S$ | $N_e * S * C_s$ | $N_e, S, C_s$ |
| Refinement (general) | $T_{\text{in}} * (1 + N_r)$ | $T * (1 + N_r)$ | $N_r * C_z$ | $N_r, C_z$ |
| Refinement (self-feedback) [150] | $T_{\text{in}} + (2T_{\text{in}} + T) * N_r$ | $T + 2T * N_r$ | – | $N_r$ |

Table 5: Token budget for representative algorithms from each meta-generation class. **Reranking.** $T_{\text{in}}$ and $T$ are the number of input tokens and output tokens for each call to the generator, respectively. For simplicity, we assume the number of input and output tokens is constant across calls to the generator. $C_s$ refers to the number of tokens required to call a scoring model (e.g., a prompted LLM) on an input and output sequence. LLM scorer refers to prompting a LLM with an input and output, and generating a scalar score (assumed to be 1 token). **Transformation.** $C_t$ refers to the number of tokens required to call a transformation function (e.g., a prompted LLM) on $N$ sequences. **Step-level search.** $T_s$ is the number of output tokens in a step, with $S$ the maximum number of steps, such that $T_s * S \geq T$. $N_b$ is the number of candidates to keep after pruning (e.g., "beam size"), and $N_e$ is the number of expansions per iteration. We assume the cost of the scorer is equal to the cost of scoring a full sequence ($C_s$). **Refinement.** $N_r$ is the number of refinement iterations. $C_z$ refers to the number of tokens required to obtain external information during a refinement iteration.

## 6.1 Token budget

Meta-generators consist of calling generators, which leads to costs associated with generating tokens. For instance, common APIs charge by the number of tokens in the input prompt and the number of output tokens. In general, meta-generators incur token costs from input tokens, output tokens, and external information.

For instance, a reranker that generates $N$ sequences incurs a cost of $T_{\text{in}} * N$ input tokens, $T * N$ output tokens, and $N * C_s$ tokens to run the scoring model, where $C_s$ is the token cost of calling the scoring model on one sequence. When the scoring model is implemented by prompting an LLM and generating a scalar quality score (assumed to cost 1 token), the external information cost is $N * (T_{\text{in}} + T + 1)$. Table 5 shows the token budget for representative algorithms from each meta-generation class.

**Step-level vs. sequence-level search.** Consider solving a mathematical problem by generating a solution that consists of multiple steps. Two strategies for doing so are (1) generating one step at a time using a step-level search algorithm, or (2) generating full solutions in a transformation or re-ranking algorithm. In this case, we can assume that $T = T_s * S$, i.e., the total number of tokens in a solution ($T$) equals the number of tokens in a step ($T_s$) times the number of steps ($S$). We can then use Table 5 to reason about when step-level search can cost fewer tokens than sequence-level search.

From Table 5, we see that step-level methods incur a cost from generating output tokens that depends on the pruning parameter $N_b$, the number of expansions per iteration $N_e$, and the number of iterations $S$. Assuming that $T_s * S = T$, step-level search has fewer output tokens than sequence-level search when $N_b * N_e < N$. For example, under these assumptions step-level beam with a beam size of 16 and 64 expansions per iteration has the same number of output tokens as best-of-1024, while lowering the expansions per iteration to 32 would be half the output token cost compared to best-of-1024.

On the other hand, Table 5 shows that step-level search calls the scoring model more often than sequence-level search methods. For instance, when $N_b * N_e = N$, step-level beam search calls the scoring model $N * S$ times compared to $N$ times with reranking. Therefore, one must also account for potential token costs associated with external information (e.g., sequence scores) when comparing meta-generator token budgets.

**Refinement vs. sequence-level search.** Similarly, we can compare the token budget for refinement versus sequence-level search. As seen in Table 5, general refinement algorithms have a lower output cost when $N_r < N$, i.e., the number of refinements is less than the $N$ in best-of-$N$. In practice this is often the case, e.g. Madaan et al. (2023) use $N_r = 3$ in many experiments, while $N$ typically ranges from 8 to 1024 in the literature. However, we need to factor in the cost of external information. For instance, when generating self-feedback as in Madaan et al. (2023), the output cost becomes $T + 2T * N_r$, meaning that 3 refinements costs $7T$ output tokens, which is still cheaper than best-of-8.

### 6.2 Scaling the token budget to improve performance

In various reasoning-related tasks such as mathematical problem solving, it has been widely observed that generation algorithms which generate multiple sequences and choose among the sequences (e.g., best-of-$N$, majority voting) can outperform generation algorithms that generate a single sequence (e.g., greedy decoding) (Cobbe et al., 2021; Wang et al., 2023b; Azerbayev et al., 2024; Lightman et al., 2024; Wang et al., 2023a; Sun et al., 2024a).

Figure 6 shows a plot from Sun et al. (2024a) that compares the relationship between the generation budget (in units of sequences) with three sequence-level approaches on the MATH500 benchmark (Lightman et al., 2024). The plot shows that these algorithms can improve monotonically by increasing the generation budget. Moreover, each algorithm has a different improvement as a function of the generation budget. For instance, at a budget of 1024 sequences, weighted voting is preferred to majority voting or best-of-N in terms of task performance. Recently, Chen et al. (2024a) found that some models can have a non-monotonic relationship between generation budget and voting performance.

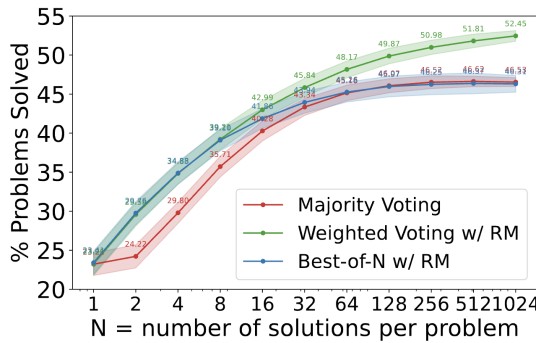

Figure 6: Plot from Sun et al. (2024a). Scaling behavior of three meta-generators in the number of samples $N$ on mathematical problem solving (MATH500).

The idea of increasing the generation budget to improve performance has appeared in many applications. For instance, AlphaCode (Li et al., 2022) generates up to a million sampled programs that are then filtered using heuristics and execution results. In theorem proving, Draft-Sketch-Prove (Jiang et al., 2023) leverage the proof checker at generation time by generating and checking many formal proof candidates, resulting in a monotonically increasing percentage of proven theorems as a function of the budget.

More formally, let $q_*(y|x) \propto 1$ if $y$ is correct, and 0 otherwise, where correctness may mean a correct solution to a mathematical problem, a valid proof, a program that passes test cases, etc. Then the goal of generation is $y_* = \arg\max_{y \in \mathcal{Y}} q_*(y|x)$. Since the space of solutions $\mathcal{Y}$ is too large, a meta-generator can approximate it by calling a generator multiple times,

$$y_* = \arg\max_{y \in \mathcal{Y}} q_*(y|x) \tag{50}$$

$$\approx \arg\max_{y^n \in y^1, \dots, y^N} q_*(y^n|x), \tag{51}$$

where $y^n \sim q(\cdot|x, p_\theta)$. It is clear that performance should improve as $N$ increases, so long as the generator $q$ assigns probability mass to correct solutions. However, in practice we do not have access to $q_*$ at test time, so different meta-generators approximate (51), e.g. with a learned verifier $v_\psi(x, y)$, or with a voting algorithm. The plot above shows that different approximations have different levels of effectiveness.

### 6.3 Minimizing the token budget

A complementary direction is minimizing the generation budget to achieve a given level of performance. One direction is to route generations to progressively more costly models. For instance, FrugalGPT (Chen et al., 2023b) first generates with a cheap model, then uses a learned scoring function to determine whether to generate again with a more expensive model, leading to significant cost reductions over calling GPT-4 in their experimental setting. Another direction is leveraging properties of specific meta-generation algorithms to reduce the number of calls. Aggarwal et al. (2023) propose to stop sampling in majority voting upon converging to a majority.

### 6.4 Compute optimal inference

When choosing or designing a meta-generator, a key consideration is the cost of the meta-generator needed to achieve a given level of performance. For example, running Monte Carlo Tree Search may give good task performance, but not if we only allow it to generate the same amount of tokens as parallel sampling. Relatedly, Kapoor et al. (2024b;a) argue that performance comparisons of meta-generation algorithms must be performed with respect to token budget and monetary cost, and that in some cases simple meta-generation baselines can provide a pareto-optimal cost-performance tradeoff compared to complex algorithms.

Wu et al. (2024) formalize these tradeoffs in terms of *compute-optimal inference*: the problem of choosing a model size, number of generated tokens, and meta-generation algorithm that minimizes error subject to a compute budget. Specifically, let the error rate $E(M, T; g)$ be a function of the number of model parameters $M$, the number of generated tokens $T$, and the generator $g$. The goal is to minimize $E$ under the constraint $c(M, T, g) = C$, where $c(M, T, g)$ is the compute used during inference, measured in floating-point operations:

$$(M_{\text{opt}}(C), T_{\text{opt}}(C), g_{\text{opt}}) = \underset{M, T, g \text{ s.t. } c(M, T; g) = C}{\arg\min} E(M, T, g), \tag{52}$$

where $M_{\text{opt}}(C)$, $T_{\text{opt}}(C)$, $g_{\text{opt}}$ denotes the choice of model size, generated tokens, and meta-generator that achieves the lowest error with compute budget $C$. Wu et al. (2024) study tradeoffs between best-of-$N$, majority voting, and tree search variants, finding that sampling more tokens from a smaller model often had better cost-performance tradeoffs compared to using a larger model at a given compute budget. Moreover, Monte-Carlo tree search often had worse cost-performance tradeoffs than the other meta-generators. Snell et al. (2024) study similar cost-performance trade-offs between best-of-$N$, step-level search, and refinement.

### 6.5 Dependence on the underlying generator(s)

The defining property of meta-generators is that they rely on calling other generation algorithms. Hence a second natural question is to what degree their performance depends on the underlying generation algorithms.

**Sampling parameters.** Chen et al. (2021) found that the optimal temperature in best-of-$N$ was dependent on $N$ for code generation with the Codex model, with higher temperatures returning better performance for higher $N$. Many prior studies use temperatures or sampling parameters that are either unexplained or ad-hoc. For instance, Minerva (Lewkowycz et al., 2022) uses majority voting with temperature 0.6 and nucleus sampling $p = 0.95$. These settings have propagated into subsequent studies (Azerbayev et al., 2024).

For some classes of meta-generators such as minimum Bayes risk (§4.2.2), the effect of sampling parameters is relatively well-studied. For example, Freitag et al. (2023) investigate the impact of the underlying sampling strategy in MBR, finding variation across strategies, with epsilon sampling performing best for translation.

## 7 Speeding up generation

In the preceding sections, we introduced generation algorithms (e.g., ancestral sampling, beam search) and meta-generation algorithms (i.e., programs involving multiple generation calls), and discussed one aspect of efficient generation: making generation cost-effective in terms of the token budget. Next we turn to another

aspect of efficiency: the speed of (meta-)generation. Speed is an inherent concern of almost any practical application of generation algorithms: users typically want outputs quickly.

Meta-generators in particular raise demands for fast generation, since they often involve generating many sequences and coordinating multiple components. For example, the meta-generators shown in Figure 6 require generating and scoring 1024 sequences. There are at least two high-level strategies one can take to speed up generation: (1) speeding up the generation of each individual sequence, and (2) leveraging structure that comes from multiple generator calls, such as shared partial outputs or the structure of the overall meta-generation program. We will consider both of these below.

Before we start, it is worth noting two points. First, the notion of "speeding up" itself needs to be made more precise and measurable. To that end we provide background on the notions of latency, throughput, and the idea that speed is often dependent on the hardware environment in which a meta-generator is run.

Second, the topics in this section are part of a rich, rapidly evolving research field that ranges from machine learning systems to programming language design. It goes without saying that our survey here merely scratches the surface. We focus our discussion on introducing key ideas, and on examining the *interaction* between the design space of (meta-)generation algorithms and generation speed.

## 7.1 Background

**Goals of speeding up generation.** Speeding up generation requires balancing between three high-level metrics: (1) **latency**, the time it takes to generate a single output; (2) **throughput**, the rate at which outputs can be produced; and (3) **quality**, measures of model quality such as loss or downstream task metrics. For instance, one might change the generation algorithm in a way that speeds up a single generation (improving latency), but removes the ability to generate outputs in parallel (degrading throughput). Other cases such as reducing the precision of model weights may improve latency and throughput, but degrade the model's task performance. Ideally, we would like to reduce latency, increase throughput, and maintain quality.

**Hardware-aware optimization.** The **underlying hardware** is a key consideration for speeding up generation. LLMs are typically run on accelerators such as GPUs or TPUs.

In the case of GPUs, performance is largely dictated by **compute** and **memory bandwidth**. Compute is typically measured via the number of floating-point operations (FLOP) used in a given operation, while memory bandwidth refers to the rate at which data can be transferred to and from memory. For example,

$$A = BC, \tag{53}$$

reads the matrices $B, C$ from memory, computes $BC$ on-chip, and writes the result out to memory. Similarly,

$$Y = \text{ReLU}(X) \tag{54}$$

must read $X$ from memory, compute $\text{ReLU}(X)$ on-chip, and write the result out to memory. However, these two operations have very different **arithmetic intensities**, defined as the ratio of compute (in FLOP) to unit of memory read or written. This results in (53), for large enough $B, C$, being **compute-bound** (bottlenecked by the rate at which operations can be performed) while (54) for large $X$ is **memory-bound** (bottlenecked by the speed of reading inputs and writing outputs to memory).

Thus, *reducing the quantity of operations performed (in FLOP) for a given step may not always proportionately transfer to an equivalent real-world speedup or cost reduction.* This is exacerbated by the properties of recent accelerators–GPUs and TPUs are heavily specialized for matrix multiplication and other high-arithmetic intensity, heavily parallelizable workloads (NVIDIA, 2017; 2020). For example, the H100 can perform up to 989.4 TFLOP/s in BF16 within a dense matrix multiplication using Tensor Cores, but only 133.8 TFLOP/s of BF16 arithmetic (NVIDIA, 2022). This specialization–and the fact that "naive" attempts to optimize performance oblivious to which operations may be the key bottlenecks may not achieve the anticipated gains–implies that **hardware-aware optimization** is a key viewpoint to take when seeking speedy generation. Algorithmic and architectural co-design with the hardware (Dao et al., 2022; Dao, 2023; Anthony et al., 2024) has yielded some of the most significant speed gains in recent years, in contrast to

| Type | Selected Examples | Strategy |
|------|-------------------|----------|
| Architectural | MQA [193], GQA [4], MLA [44], $\cdots$ 
 RWKV [174], Mamba [77], $\cdots$ | Efficient attention 
 Transformer alternative |
| Compression | GPTQ [60], AWQ [135], SqueezeLLM [106] , $\cdots$ 
 LLM.int8() [46], Smoothquant [230], QuaRot [7], $\cdots$ 
 FlexGen [194], KVQuant [88], W4A8KV4 [136], $\cdots$ | Quantize weights 
 Quantize activations 
 Quantize KV Cache |
| Hardware-aware impl. | Rabe & Staats (2022), FlashAttention [41; 40], $\cdots$ 
 Triton [212], Torch compile [180, Cutlass [210], $\cdots$ | Efficient attention 
 Libraries/tooling |
| Parallelize over time | Speculative decoding [123; 26], SpecInfer [157], $\cdots$ | Draft-then-verify |

Table 6: Outline of classes of techniques for speeding up a single generation call.

approaches seeking to minimize theoretical complexity that are disconnected from the hardware level. On the flip side, however, Hooker (2020) discuss the notion of the *hardware lottery*–the idea that co-design of novel techniques creates adverse selection effects, where research ideas "off the beaten path" are dispreferred because they interact less well with existing hardware.

## 7.2   Speeding up the generator

Generation algorithms with autoregressive language models depend on computing next-token distributions. Given an input sequence $(y_{<t}, x)$, typical implementations start with an initial "prefill" step that computes

$$p_\theta(\cdot | y_{<t}, x) = \text{softmax}(s_\theta(\cdot | y_{<t}, x)). \tag{55}$$

Performing this step returns *two outputs*: $p_\theta(\cdot | y_{<t}, x)$, the probability distribution over immediate next tokens following $(y_{<t}, x)$ that we have discussed previously, and a "state" $S_{y_{<t}, x}$ created as a byproduct of processing $y_{<t}, x$. For a Transformer (Vaswani et al., 2017) $S_{y_{<t}, x}$ is produced by retaining all keys and values from timesteps up to $y_{t-1}$ within the attention for each layer.[3] This is termed the "Key-Value (KV) Cache" produced by attention at each layer. At this step, we may sample a next-token $y_t$ from $p_\theta(\cdot | y_{<t}, x)$.

Subsequently, to generate additional new tokens we may perform any number of "decoding" steps, where each step selects a token from a next-token distribution. For example, the $t+1$'th step selects a token using:

$$p_\theta(\cdot | y_{<t+1}, x, S_{y_{\leq t}, x}) = \text{softmax}(s_\theta(\cdot | y_{<t+1}, x, S_{y_{\leq t}, x})). \tag{56}$$

Here, we feed the state $S_{y_{\leq t}, x}$ into the model, representing the already-processed sequence. Each decoding step saves on computations that are cached in the state, such as the attention keys and values from the preceding steps. After selecting the $t + 1$'th token, the state is updated to $S_{y_{\leq t+1}, x}$. These decoding steps may be repeated until we have finished generating a sequence.

One can accelerate a single generation from an LM by speeding up the time taken per step, such as through architectural modifications, model compression, hardware-aware implementation decisions, or by clever parallelization during autoregressive generation. We discuss each of these in the following paragraphs.

**Architectural modifications.**   One strategy is to modify the model architecture. For example, multi-query (Shazeer, 2019) and grouped-query (Ainslie et al., 2023a) attention propose the use of fewer key and value heads in transformers' attention, leading to reduced KV Cache sizes. Smaller KV Cache sizes can lower memory bandwidth demands, or provide the ability to process larger batches concurrently at a time by enabling more requests to be stored in GPU memory. Similarly, DeepSeek-AI (2024) propose multi-headed latent attention, attempting to retain the reduced KV Cache of GQA while improving model quality. The $O(t^2)$ complexity of attention ($O(t)$ for each decoding step) can slow generation down as sequences become longer, so another option is to forego the transformer architecture or its attention layer altogether. For example, traditional recurrent language models (Elman, 1990; Mikolov et al., 2010) compute a next-token distribution by maintaining a hidden state, leading to a $O(t)$ time and space complexity ($O(1)$ per step).

---

[3]The core attention operation is softmax $\left(QK^T/\sqrt{d}\right) V$, where $Q, K, V \in \mathbb{R}^{t \times d}$ are referred to as queries, keys, and values, respectively, $t$ is the time dimension, and $d$ is the hidden dimension.

Recent architectures draw on ideas from recurrent language models (Hutchins et al., 2022; Peng et al., 2023; De et al., 2024; Yang et al., 2024) and/or state-space models (Gu & Dao, 2023; Lieber et al., 2024) to achieve sub-quadratic time and space complexities. Although models can occasionally be adapted post-hoc from a transformer architecture to one of these more efficient variants (Zhang et al., 2024; Ainslie et al., 2023a), this adaptation can degrade model quality or require substantial compute.

**Model compression.**  Adjacent to architectural modifications, one can *compress* a model into a more efficient form after the fact. *Distillation* can transfer knowledge from a more capable teacher model into a smaller one (Hinton et al., 2015; Sanh et al., 2020), or models can be *quantized* to reduce the floating-point precision of the model's weights which reduces the memory footprint of the model and in turn speeds up generation in memory bandwidth-constrained settings (Dettmers et al. (2022); Frantar et al. (2023); Dettmers et al. (2023); PyTorch (2023), *inter alia*). Model activations can also be quantized (Ashkboos et al., 2024; Xiao et al., 2024a; Lin et al., 2024b). Approaches to sparsify or prune model weights (Frantar & Alistarh (2023), *inter alia*) can also be used. Such compression approaches frequently, but not always, degrade performance and require training to perform or to recover performance on a limited distribution.

**Hardware-aware implementation.**  A number of optimizations may be performed without modifying the model architecture or *what* operations must be performed, simply *how* they are performed.

For instance, Flash Attention (Dao et al., 2022; Dao, 2023) famously overcomes the $O(t^2)$ space complexity of self-attention by adapting the algorithm proposed by Rabe & Staats (2022) for computing self-attention based on online softmax (Milakov & Gimelshein, 2018; Jang et al., 2019) and blockwise computation, crucially without changing the output of the attention mechanism, simply its mapping to hardware. Similarly, Flash Decoding (Dao et al., 2023) accelerates the attention operation during decoding by adding extra parallelism over the sequence dimension, allowing the GPU to be fully saturated even for small query and batch sizes, but only changing the order and mapping of operations on-device, not the end result (up to numeric precision).

Numerous software tools (Tillet et al. (2019); PyTorch (2023); Thakkar et al. (2023), *inter alia*) can enable fast decoding and efficient low-level implementation in practice. Overall, while architectural modifications to the model itself can increase the *ceiling* on generation speed, effective *implementation* is key for achieving performance anywhere near this ceiling on current accelerators.

**Parallelization across time.**  Rather than speeding up the core next-token operation, the *draft-then-verify* (also called "speculative sampling" or "speculative decoding") pattern leverages clever parallelization during autoregressive generation. Draft-then-verify consists of generating proposed next-tokens with a fast method (e.g., a smaller model), computing next-token distributions given the proposed tokens *in parallel*, and either keeping or rejecting the proposed tokens.

For example, previously we briefly referred to speculative sampling (Leviathan et al., 2022; Chen et al., 2023a). This method assumes a language model $p_\theta(y_t|y_{<t})$ and an efficient *draft* model $q(y_t|y_{<t})$. At a given step $t$, it generates a continuation $y_t, y_{t+1}, \ldots, y_{t+k}$ using $q$, then computes the next token distributions $p_\theta(y_t|y_{<t}), \ldots, p_\theta(y_{t+k}|y_{<t+k})$ in parallel. Finally, it processes each proposed token, keeping it if $q(y_{t'}|y_{<t'}) \le p_\theta(y_{t'}|y_{<t'})$, and rejecting it when $q(y_{t'}|y_{<t'}) > p_\theta(y_{t'}|y_{<t'})$ with probability $p_\theta(y_{t'}|y_{<t'})/q(y_{t'}|y_{<t'})$ or if a preceding token was rejected. Intuitively, as long as (i) generating with $q()$ is much faster than computing the distributions with $p_\theta$ in sequence, and (ii) the rejection rate is not too high, then speculative sampling will speed up generation without affecting the original model's output distribution or quality.

Several methods iterate on ideas underlying speculative sampling, including guessing and verifying a tree of proposed tokens (Miao et al., 2023b; Sun et al., 2024b; Chen et al., 2024c), using alternative proposal models $q$ (Miao et al., 2023b; Cai et al., 2024), using prompt n-grams as proposals (Yang et al., 2023b), or generating in parallel and reusing the generated n-grams as proposals (Fu et al., 2024). Interestingly, many speculative sampling approaches which require a separate draft model $q()$ require *more* total FLOP in order to generate a given sequence (Chen et al., 2023a; Leviathan et al., 2023; Fu et al., 2024). However, because the decoding step is typically memory-bound, the increased parallelism afforded is more than sufficient to provide substantial generation speedups.

| Type | Selected Examples |
|------|-------------------|
| State reuse | PagedAttention memory sharing [118], RadixAttention [252] |
| State compression | Gisting [160], KV Cache compression [136; 251] |
| Improved batching | Continuous batching [241], Disaggregated prefill [253] |
| Program-level optimization | GPT-4 graph rewriting [252], DSPy [103] |

Table 7: Outline of techniques for speeding up meta-generation algorithms, requiring many calls to an underlying generator with often-predictable traffic patterns. Refer to the main text for more examples.

### 7.3 Speeding up meta-generation algorithms

While in §7.2 we note approaches to speeding up a *single* autoregressive generation call, the space of possible optimizations is larger when considering usage patterns such as those found in meta-generation algorithms, where multiple or many calls are made to the same model over time, often in a predictable way.

#### 7.3.1 Leveraging shared prefixes.

The repeated model generation calls that occur in meta-generation algorithms crucially often share similarities in input. Most importantly, they often share *prefixes* across generation calls. This provides an opportunity to save on computation and dramatically speed up generation throughput.

**KV Cache and state reuse.** In typical transformers, because the KV Cache is updated by appending the keys and values of a new token to the cache, the KV Cache for shared prefixes can be "prefilled" only a single time and reused across generation calls that share this input prefix. For example, in parallel meta-generation algorithms (§4.2) such as Best-of-$N$, when producing an $N$-best list $\{y^{(n)}\}_{n=1}^N \sim g$, generating each candidate $y^i$ requires a "prefill" step computing $S_x$ in order to sample $y_1^i$ and each successive token in $y^i$. Simply computing $S_x$ once and reusing it when sampling each $y^i$ can save significant computation and time, in effect reducing the input token count for such algorithms by a factor of $N$ (Table 5).

Making such state sharing efficient requires careful handling of the state in memory, but can significantly speed up throughput by allowing more outputs to be processed at a time as a result of lightened GPU memory requirements (Kwon et al., 2023). It can also be generalized beyond a single prefix being shared (Zheng et al., 2023) in order to handle branching, complex trees of already-processed inputs. Later work has shown that redundant computation can be eliminated even further, allowing specific speed optimizations in the presence of shared prefixes (Juravsky et al., 2024).

**KV Cache and state compression.** A complementary line of work approaches the challenge of handling reused model states or KV Caches efficiently by *compressing* them, reducing the storage required. Gisting (Mu et al., 2023) and other related techniques (Chevalier et al., 2023; Ge et al., 2024b) tackle the sub-problem of *long, frequently-recurring input prompts* by learning to produce a series of "soft" tokens (trained token embeddings) which compress a given large input prompt into a much smaller, more compact state. These methods can be viewed as a generalization of prefix tuning or prompt tuning (Li & Liang, 2021; Lester et al., 2021). Other methods explore the shortening of KV Caches via determining which items to retain or evict from the input prompt, or at each step whether to append new keys and values to the cache (Ge et al., 2024a; Liu et al., 2023; Zhang et al., 2023; Li et al., 2024b; Nawrot et al., 2024; Raposo et al., 2024; Xiao et al., 2024b). Much like model weights, the KV Cache can also be compressed via reducing its storage precision, such as via quantization (Sheng et al., 2023; Lin et al., 2024b; Zhao et al., 2024c; Zirui Liu et al., 2023). Thus, the memory bandwidth cost of loading the KV Cache from memory is reduced, and more tokens' caches can be fit onto GPU memory. However, these compression techniques can lose model quality.

#### 7.3.2 Optimizing computational graphs

Finally, a class of optimizations takes into consideration the programmatic structure of the meta-generator.

**Caching.** Caching model state across calls to a generator as done by Zheng et al. (2023); Juravsky et al. (2024); Kwon et al. (2023) and discussed previously can be beneficial for algorithms that involve backtracking (e.g., tree search), or in general programs that involve duplicate generator calls.

**Graph optimization.** Additionally, the computational graph of such programs can be optimized and rewritten with efficiency in mind, by hand or automatically. For example, SGLang uses GPT-4 in its optimization of programs to reorder computational graph nodes (Zheng et al., 2023), and DSPy optimizes performance or cost of LM programs via automating prompt engineering (Khattab et al., 2024).

**Algorithm-specific optimization.** When the specific algorithm is known, optimizations can be made even more targeted, such as speeding up voting algorithms by stopping early upon converging to a majority (Aggarwal et al., 2023), or a host of methods that optimize MBR-style algorithms, including confidence-based hypothesis pruning (Cheng & Vlachos, 2023) or leveraging reductions of MBR (Jinnai & Ariu, 2024).

### 7.4 Libraries and tools for fast generation

We briefly note a few useful libraries and tools for fast and efficient generation, although the space of useful tools and libraries is in particular especially subject to fast change. vLLM (Kwon et al., 2023) is a highly popular library that introduced PagedAttention and implements a number of up-to-date optimizations for fast generation, including continuous batching, prefix caching and reuse, various model and KV cache quantization techniques, speculative decoding, and more. TensorRT-LLM is another highly efficient LLM serving library. Especially relevant to this survey, SGLang (Zheng et al., 2023) builds on vLLM to provide a domain-specific language optimized for meta-generation.

GPT-Fast (PyTorch, 2023) provides a minimal implementation of latency-constrained fast decoding in PyTorch, and is designed to be useful for prototyping new ideas and to demonstrate the ease of optimizing low-latency unbatched decoding workloads using simple tools such as `torch.compile`.

For end users, especially those without easy access to data center-grade or high-end consumer-grade GPUs, a number of libraries also implement fast decoding on CPU, which presents its own set of challenges not fully explored in this paper. Libraries such as Llama.cpp[4] are popular for consumers, and libraries such as DeepSpeed-Inference (Aminabadi et al., 2022) or PowerInfer (Song et al., 2023; Xue et al., 2024) explore optimizations when *offloading* activations or parameters to slower-access storage or CPU RAM, which require systems considerations beyond those discussed for the more typical homogenous accelerator setting.

## 8 Discussion: why use sophisticated generation algorithms?

Finally, we return to the question that we posed in the introduction: *why are sophisticated generation algorithms needed at all?* For example, we might imagine that simply sampling once from the model's unmodified output distribution, $y \sim p_\theta(y|x)$ is sufficient. We offer some takeaways based on our survey.

**Takeaway 1: iron out degeneracies in the learned distribution.** In §3.2 and §3.3 we discussed introducing token-level adapter algorithms to avoid errors in the model's distribution (for instance, when the learned model assigns too much probability to sequences that have low or zero probability under the true distribution). At a qualitative level, examples encountered in practice include ancestral sampling resulting in incoherent sequences. At another extreme, MAP decoding algorithms (§3.1) can result in unnaturally repetitive sequences that are nevertheless assigned high probability by the model, or even empty sequences. These degeneracies motivate the use of generation algorithms with alternative goals, such as minimizing Bayes Risk (§4.2.2), or the use of a token-level adapter. In these cases, generation algorithms offer a mechanism for modifying the resulting generation distribution to remove these errors.

*Moving forward.* As models improve, will generation algorithms for these cases still be needed? Since the aforementioned errors stem from a model imperfection, it is plausible that future models will not have these imperfections. Moreover, taking simplicity of the overall generation system as an objective to strive for, we might explicitly strive to ultimately remove the generation algorithm for these cases. On the other hand,

---

[4]https://github.com/ggerganov/llama.cpp

imperfections may come from subtle design choices, such as the use of softmax (Finlayson et al., 2024a), or the choice of an autoregressive architecture. Therefore, we speculate that on the way to achieving this objective, it will remain important to identify degeneracies in existing models, introduce practical methods to mitigate them, and ultimately gain an understanding that can be used in the design of new methods.

**Takeaway 2: align the generation distribution with a new objective.** Above we discussed how the learned distribution $p_\theta$ may not equal the desired distribution of generations $q_*$ (e.g., §2.2.3, §3.4). For instance, language modeling corpora often contain sequences considered offensive in many contexts (Gehman et al., 2020), and we may want to generate only non-offensive outputs. We have seen examples of generation algorithms that reweight the model's distribution using another model (e.g., one trained to adjust the distribution so that it optimizes a non-toxicity reward, §3.4), or draw a large number of samples from the model and select one with a reward function (a form of rejection sampling, §4.2.3). In these cases, the generation algorithm offers a layer of "control" over the generation distribution, allowing us to shift the generation distribution to a desired one.

*Moving forward.* Moving forward, we speculate that the language model's learned distribution will not always align with the desired distribution for all possible uses. Thus, using a generation algorithm to shift the model's distribution to a desired one may withstand the test of time. On the other hand, previously we discussed the connection between generation algorithms and reinforcement learning (e.g., see refer to the discussion of Equations 19, 29). Indeed, if we have a reward function, then for a particular application a model may be finetuned to match the distribution induced by the reward function, offering a potential channel to removing the complexity associated with generation algorithms.

In the near term, users may interact with models through black-box APIs that are simply imperfect for their application of interest, but obtaining a filtering function or scoring model is relatively straightforward. Thus we speculate that in practical cases, users will continue to benefit from algorithms such as rejection sampling (§4.2.3) that shift the generation distribution using repeated calls to the generator model.

**Takeaway 3: dynamic computation.** In §4, we discussed how generation algorithms can be viewed as searching through the output space for a desired sequence (§4.3), and that generation algorithms offer a mechanism for using compute to expand the coverage of states explored during the search. For example, best-of-N algorithms (Equation 5) use compute to search for a solution by generating $N$ hypotheses. More formally, we can write down the objective of many algorithms as approximating the maximization of some scoring function, and doing so with a sampling-based approximation indeed results in a better approximate solution as the number of samples increases (e.g., see Equations 30, 50). We saw an example (Figure 6) where taking this approach was effective for increasing the probability of a generator's output being correct in reasoning problems. In this sense, generation algorithms offer a mechanism for dynamically expending compute with a language model to improve performance.

*Moving forward.* Moving forward, we see several cases. For some tasks, models may become so good that a single sequence is all that is needed to arrive at a desired state with near 100% probability. In these cases, the generation algorithm may not be useful. In challenging cases, the model may have acquired a useful representation, but may benefit from exploring the output space through backtracking, revision, etc. before arriving at a final solution. In these cases, the computation expended at inference time remains useful. Nevertheless, there are at least two potential alternatives to generation algorithms with respect to dynamic computation. One is building dynamic computation into the architecture (Raposo et al., 2024) or learning algorithm (Goyal et al., 2024; Zelikman et al., 2024a), so that allocating more compute to harder problems is automatically handled by the model. A second alternative is *learning the search algorithm* (Chang et al., 2015; Lehnert et al., 2024; Gandhi et al., 2024), then using a vanilla generation algorithm.

In the nearer term, at the application level users may interact with models through black-box APIs that are simply imperfect. Thus we speculate that users will continue to benefit from using the API as a hypothesis generator that is called multiple times within a search algorithm, rather than a model that is called once for an answer. From a research perspective, the relationship between generation-time compute and performance requires further investigation. For instance, we saw above that the relationship varies by generation algorithm

in voting settings (§6.2). In general, designing algorithms that optimally leverage test-time compute and studying their trade-offs requires further research (Chen et al., 2024a; Snell et al., 2024; Wu et al., 2024).

Finally, meta-generators can be seen as a new kind of computational graph that itself can be optimized, such as with architecture search (Saad-Falcon et al., 2024) or even with a meta-generator (e.g., refinement (Yuksekgonul et al., 2024)). These emerging perspectives open many future directions to explore.

**Takeaway 4: leveraging external information.** In §5 we saw several ways in which generation algorithms incorporate information that is external to the language model. This includes predictions from other models, prompts, external tools or verifiers, or generally inputs and outputs from an external environment.

*Moving forward.* Moving forward, we may expect that a generation algorithm that has information beyond that present in a language model may exceed the performance of the language model on its own. We speculate that language models on their own may either be unable to solve subproblems that are required for a complete generation (e.g., performing an algebraic computation that is simple for a computer algebra system, or retrieving a piece of information that is outside of the model's parametric knowledge), or that doing so is an inefficient allocation of computation. Moreover, we expect that in many challenging settings, it may be necessary to decompose problems into a cascade of generations, interact with an environment, and iteratively arrive at a final generation. In all of these cases, generation algorithms that incorporate external information, be it in prompts that alter a model's distribution within a chain (§4.1) or environmental feedback (§4.4, §5.2), offer a range of possibilities to explore in future research.

Finally, we expect that discrete autoregressive sequence generators will increasingly be used in domains traditionally outside of those considered in natural language processing, such as a component in an agent that receives states and takes actions in a potentially stochastic environment. In these cases, using external information is inherent to the problem. How this translates to generation algorithms remains an open area of research. For instance, notions of planning (e.g., §4.3, §5.2) are natural fits for planning actions in interactive environments. In the near term, we expect to see generation algorithms developed in the context of language generation such as tree search and refinement (§4.4) integrated into these settings.

**Takeaway 5: speed up generation.** Finally, in §7 we saw examples of how generation algorithms can themselves be used to speed up generation, even in the case of ancestral sampling from a language model (§7.2). On the other hand, the increasingly diverse configurations of meta-generation algorithms present new demands and opportunities for fast generation (§7.3).

*Moving forward.* Regardless of the future form of sequence generators, we expect that there will always be a need and opportunity for speeding up generation. Naturally, one must consider the expected marginal benefit of developing a new method for speeding up generation compared to existing methods. Given the evolving nature of model scale, architectures, applications, and compute environments, we expect the marginal benefits of new methods to remain high for the foreseeable future. In particular, optimized generation algorithms that involve multiple models, external information, and/or cascaded generation is a nascent research area.

## 9 Conclusion

We surveyed generation algorithms for language models. We motivated generation algorithms, formalized their goals, and provided a unified treatment of three themes: token-level generation algorithms, meta-generation algorithms, and efficient generation. Our survey brings together past research from the decoding, LLM reasoning, and machine learning systems communities, and identifies directions for future work.

**Acknowledgments**

We thank Akari Asai, Rob Brekelmans, Yejin Choi, Saibo Geng, Konstantin Golobokov, Shibo Hao, Omar Khattab, Taehyeon Kim, Chufan Shi, Aditi Raghunathan, Sasha Rush, Peter West, Mert Yuksekgonul, and the TMLR reviewers for helpful discussions. This work was partially supported by NSF DMS-2134012, CCF-2019844, IARPA 2022-22072200003. SW would like to thank Convergent Research. Part of this work was done while Z. Harchaoui was visiting the Simons Institute for the Theory of Computing.

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
