# OpenReview forum: "From Decoding to Meta-Generation: Inference-time Algorithms for Large Language Models"
_TMLR — Accepted by TMLR_

### Review · Reviewer_YVip · 2024-07-27

**Summary Of Contributions:**

This paper presents a survey of inference time algorithms for using large language models. These inference time algorithms make use of information already existing in the LLM, and apply this information to make inference for a target distribution.

This survey presents a survey from three aspects, including:
- generation algorithms: these are the most standard generation procedures of a trained language model, such as the generation of the next token.

- meta-generation algorithms: these are procedures that make use of another generation algorithm during its execution. Examples include models that can output Python code or execute Python code.

- efficient generation: these algorithms discuss the computational cost of the generation procedure, focusing on the inference time.

For each aspect, this paper presents a very detailed discussion, starting from the mathematical definition of the algorithms; then, a list of detailed references and implementations of that definition.

Finally, this paper also presents a thoughtful discussion of the takeaways from this survey. These discussions help highlight avenues that could be promising for future work.

**Audience:**

Yes

**Broader Impact Concerns:**

I do not have any concerns with regard to Broader Impacts. I think this survey paper should be able to generate further discussions and interest regarding generation algorithms in language models.

**Claims And Evidence:**

Yes

**Requested Changes:**

- At various section titles, the paper has an extra dot. Examples include sections 4.3, 6.2, 6.3, 7.3.1, 7.3.2, 7.4. Possibly this is due to formatting changes. So please double check to make sure.

- There are duplicate references in the reference list. See, e.g., the bottom of page 37. In addition, some references contain the editors of the conferences cited, whereas others do not. Please cross check to ensure consistency.

- I feel like there are a lot of terminologies in the survey (possibly due to an extensive number of topics covered). This sometimes had the opposite effect on the reader, as they have to grapple with all the terms. For example, in Table 1, there are over 10 sampling methods. Some of them are clear to the reader but many of many may not be. It would be helpful to add some pointers to the terminology, so at least the reader knows where to look for the definition/citation of the method if they want.

    - What does "extrinsic" mean in Table 1?

- The term "Adapter" and "Adaptor" are both used in various places. Please cross check to ensure consistency.

- For the mathematical definitions of each algorithm, I'm having difficulty with catching up on the notations---For instance, in Definition 4, what is $\phi$, and why does it come into the left hand side?

    - In Equation (8), what does the hat above $y$ refer to? and why does the previous Equation (7) not use this hat?

    - In Equation (10), does the approximation $\approx$ actually hold, or is it just a metaphor of the greedy decoding?

- In Definition 2, I don't understand the meaning of this Equation (5). Please elaborate. Similarly, please elaborate on Equation (6).

**Strengths And Weaknesses:**

Strengths:

- This survey provides a list of most up-to-date references on a very relevant topic of language models. The references and the accompanying materials could serve as self-study materials for researchers interested in developments in this field.

- The three categories of generation procedures provide a good coverage of ongoing developments in this field, which would be of interest to NLP, ML systems, and LLMs.

- The paper provides very detailed discussions of the key techniques.

    - For generation algorithms, this paper starts by presenting decoding and sampling schemes. The reader is reminded of these commonly used techniques, such as greedy decoding, ancestral sampling.

    - For meta-generation algorithm, the paper talks about best-of-N ranking methods, beam search.

    - For efficient generation, the paper provides a discussion about various reasoning related applications, architectural modifications, optimizing computational graphs, systems level improvement, etc.

Weaknesses:

- I think it would be useful to discuss other existing surveys in this space (broadly defined). Examples include, but may not be limited to, foundation models, data selection, transfer learning for NLP. I feel like placing this survey inside a range of other existing surveys will better serve the purpose of readers, helping them nagivate these online materials for understanding the research space.

- I think it would also be helpful to discuss some limitations of this survey, including things like what topics are not covered by the survey. Again, this will help better serve the purpose, clearly specifying the scope of the paper.

---

> ### Author Response · Authors · 2024-09-19
> **Author response**
>
> ## Author response
>
> Thank you for your review and your feedback. We appreciate the positive comments on the detail, thoughtfulness, and coverage of our survey.
>
> Please find our responses to your feedback below. We have also integrated the changes that you requested into the paper.
>
> ### Comment 1: (other surveys)
> > I think it would be useful to discuss other existing surveys in this space (broadly defined). Examples include, but may not be limited to, foundation models, data selection, transfer learning for NLP. I feel like placing this survey inside a range of other existing surveys will better serve the purpose of readers, helping them nagivate these online materials for understanding the research space.
>
> Thank you for pointing this out. In the new revision (Introduction section) we added a paragraph about existing surveys as well as the unique angle and need for our survey.
>
>
>
> ### Comment 2: (limitations/topics not covered in the survey)
> > I think it would also be helpful to discuss some limitations of this survey, including things like what topics are not covered by the survey. Again, this will help better serve the purpose, clearly specifying the scope of the paper.
>
> In the revised version we have incorporated some commentary into the paragraph about existing surveys. For instance, we focus on autoregressive generation but do not cover non-autoregressive generation.
>
> ### Request 1: (formatting, spelling)
> > - At various section titles, the paper has an extra dot. Examples include sections 4.3, 6.2, 6.3, 7.3.1, 7.3.2, 7.4. Possibly this is due to formatting changes. So please double check to make sure.
> > - There are duplicate references in the reference list. See, e.g., the bottom of page 37. In addition, some references contain the editors of the conferences cited, whereas others do not. Please cross check to ensure consistency.
> > - The term "Adapter" and "Adaptor" are both used in various places. Please cross check to ensure consistency.
>
> Thank you for these suggestions, we have integrated them into the revised version of the paper.
>
> ### Request 2: (terminology)
> > I feel like there are a lot of terminologies in the survey (possibly due to an extensive number of topics covered). This sometimes had the opposite effect on the reader, as they have to grapple with all the terms. For example, in Table 1, there are over 10 sampling methods. Some of them are clear to the reader but many of many may not be. It would be helpful to add some pointers to the terminology, so at least the reader knows where to look for the definition/citation of the method if they want. What does "extrinsic" mean in Table 1?
>
> Thank you for your feedback on these points. We have added a summary of common notation that is used in the survey in the preliminaries section (Table 1). We hope that this is a useful reference while reading through the survey.
>
> Thanks for pointing out the ambiguity in Table 1 (which is now Table 2). We have added citations next to each method, and clarified the meaning of extrinsic in the caption (a model or solver separate from the underlying language model).
>
> ### Request 3: (notation, specific requests)
> > For the mathematical definitions of each algorithm, I'm having difficulty with catching up on the notations---
> - For instance, in Definition 4, what is ϕ, and why does it come into the left hand side?
> - In Equation (8), what does the hat above y refer to? and why does the previous Equation (7) not use this hat?
> - In Equation (10), does the approximation ≈ actually hold, or is it just a metaphor of the greedy decoding?
> - In Definition 2, I don't understand the meaning of this Equation (5). Please elaborate. Similarly, please elaborate on Equation (6).
>
> We’ve added a notation table (Table 1) which will hopefully resolve some of these. Here are specific answers to your questions:
> - $\phi$ refers to any additional parameters to a generator
> - $\hat y_t$ is used to denote a generated token
> - The approximation in (10) does hold; i.e. greedy decoding is an approximate MAP decoding algorithm
> - Thanks for pointing out the ambiguity in Definition 2; previously $q_p$ and $q$ were not explained. We have clarified Definition 2 in the revision.
> - Equation 6 formalizes the idea of generation algorithms that are designed to sample from a target distribution. That is, the generation distribution $q$ is close to the target distribution $q_*$ in terms of a statistical divergence, and the generation algorithm provides an unbiased sample from the target distribution (in the ideal case).

---

### Review · Reviewer_cvxc · 2024-08-12

**Summary Of Contributions:**

The paper provides a comprehensive survey of Generating high-quality samples efficiently from a large language model (LLM). It categorizes the research directions into three main topics: token-level generation methods, meta-generation methods, and efficient generation methods. The overall goal of the paper is to provide a unified framework for the research directions, summarize overall trends, and suggest future directions.

The paper first comprehensively discusses a survey of token-level generation algorithms under a unified formalism. Then, it introduces the concept of meta-generation algorithms, including all algorithms that use "sub-generators," and further categorizes them into chained, parallel, step-level, and refinement-based meta-generators. For efficiency and speed considerations, the papers provide an analysis of theoretical token costs for generators and meta-generators and then discuss various techniques to minimize the costs and speed up the generation, including both hardware and software aspects.

Finally, the paper discusses its takeaways and future directions related to the surveyed methods. It suggests that research should improve the [token-level] generation method to simplify the way we tackle degeneracies. The paper, however, suggests that sophisticated generation algorithms will continue to be necessary for aligning the model’s distribution with specific user objectives and leveraging external information. In addition, the research should continue to work on dynamic computation or other speed-up methods of LLMs to reduce cost and time as the model scales will continue to rise.

**Audience:**

Yes

**Broader Impact Concerns:**

No ethical concerns.

**Claims And Evidence:**

Yes

**Requested Changes:**

1. The paper is helpful and unique, but it might need additional justification. The authors should discuss existing surveys to motivate this survey paper and the proposed unified formalism.  Some existing surveys include:

    - https://www.ijcai.org/proceedings/2021/612
    - https://arxiv.org/abs/1803.07133
    - https://arxiv.org/abs/2303.18223

2. Provide a reference of notations. This could be in an appendix of the paper. It would also help to provide a section summary to aid.
3. Mentioning specific categories (e.g., sub-sections) in Section 8 will help clarify and strengthen the takeaways.

**Strengths And Weaknesses:**

### Strengths

1. The paper offers a thorough survey of existing generation algorithms, categorizing and explaining their functionalities, motivations, and impact on language model performance.
2. The paper introduces a framework for understanding and categorizing meta-generation algorithms. This helps provide some structures to compare the current "prompt-engineering" research.
3. The paper provides explicit, practical insights into token cost analysis and reviews tools and libraries that facilitate these improvements.

### Weaknesses

1. The paper does not discuss the existing surveys nor motivate the need for a new survey or the proposed unified formalism with existing literature.
2. The formalism helps clarify the paper but makes it hard to read and follow. The paper is rather long and introduces many notations that readers might have to come back to look up often.
3. Section 8 often discusses "generation" methods in general, leaving some ambiguity between token-level generations and meta-generations. Thus, the suggested future research directions are somewhat unclear.

---

> ### Author Response · Authors · 2024-09-19
> **Author response**
>
> ## Author response
>
> Thank you for your review and your feedback. We appreciate the positive comments about the paper’s thoroughness, helpful framework, and practical insights.
>
> Please find our responses to your feedback below. We have also integrated the changes that you requested into the paper.
>
> ### Comment/Request 1: (existing surveys and motivation)
> >The paper does not discuss the existing surveys nor motivate the need for a new survey or the proposed unified formalism with existing literature.
> >The paper is helpful and unique, but it might need additional justification. The authors should discuss existing surveys to motivate this survey paper and the proposed unified formalism. Some existing surveys include:
> https://www.ijcai.org/proceedings/2021/612
> https://arxiv.org/abs/1803.07133
> https://arxiv.org/abs/2303.18223
>
> Thank you for pointing this out. In the new revision (Introduction section) we added a paragraph about existing surveys as well as the unique angle and need for our survey.
>
> ### Comment/Request 2: (notations/formalism)
> > The formalism helps clarify the paper but makes it hard to read and follow. The paper is rather long and introduces many notations that readers might have to come back to look up often.
> > Provide a reference of notations. This could be in an appendix of the paper. It would also help to provide a section summary to aid.
>
> Thank you for your suggestion on adding a reference for notations. We have added a summary table to the preliminaries section (Table 1) to address this.
>
> ### Comment/request 3: (clarification of methods in future research)
> > Section 8 often discusses "generation" methods in general, leaving some ambiguity between token-level generations and meta-generations. Thus, the suggested future research directions are somewhat unclear.
> > Mentioning specific categories (e.g., sub-sections) in Section 8 will help clarify and strengthen the takeaways.
>
> Thank you for this great suggestion. We have added mentions of the specific sub-sections (and some equations) in section 8 to help with the presentation of the takeaways.

---

### Review · Reviewer_zXpT · 2024-09-06

**Summary Of Contributions:**

The paper presents a holistic discussion of inference-time algorithms, ranging from token-level generation to meta-generation methods. The purposes and pitfalls of various decoding methods are covered in the first part of the paper. Then, treating LMs as black boxes, the authors shift focus to meta-generation, summarizing ways to improve generation quality by utilizing auxiliary components and external information. Efficient generation schemes that employ parallelism, compression, caching, etc. are presented as means to speed up the generation process. The paper eventually concludes with five key takeaways as to why, in certain cases, sampling from an original output distribution could be insufficient and inference-time algorithms are needed.

**Audience:**

Yes

**Claims And Evidence:**

Yes

**Requested Changes:**

- Fig. 2, consider changing candidate tokens to more probable next tokens to simulate real-life cases.
- Page 24, footnote, mixed usage of notation $T$ for matrix transpose and the time dimension.

Recommended
- Section 4.2.3, consider supplying the analytical KL expression for BoN $log(n)-\frac{n-1}{n}$ from
Stiennon et al., "Learning to summarize from human feedback", In Proceedings of the 34th International Conference on Neural Information Processing Systems, NIPS’20
- Section 4.3, consider citing
Liu et al., "Don't throw away your value model! Generating more preferable text with Value-Guided Monte-Carlo Tree Search decoding", arXiv, 2023

**Strengths And Weaknesses:**

Strengths:
- To the best of my knowledge, the paper is exhaustive in its coverage of previous inference-time methods. When it comes to a fine-grained category, the previous works selected and discussed in detail are the most impactful ones from a line of qualified works.
- The terms are well-defined and the use of mathematical expressions is accurate.
- The paper is rich in its presentation of key properties related to certain methods.

Concerns:
- The three areas—generation algorithms, meta-generation, and efficient generation—are presented in parallel, but efficient generation seems to be loosely connected to the first two at first sight: while generation algorithms and meta-generation methods alter the generation process to yield desirable outputs, efficient generation is about reducing cost and improving generation speed. In section 7, the authors touch on how certain efficient techniques could improve aforementioned inference-time methods which evinces its interconnections with the first two topics. I would suggest authors modify the narrative of the abstract and introduction so that efficient generation is not a stand-alone area parallel to the other two, but more like a separate subject that can be applied to the first two areas.

- The paper lacks analysis of different inference-time methods’ performance, especially a quantitative one.
  - It is understandable for a survey paper to be as general as possible to apply to as many cases and scenarios, but it does not hurt to pick a single downstream application/task to offer insights into the uniqueness of each method. For example, authors could elaborate on how certain algorithms behave in NMT tasks.
  - Readers may wonder “When I have excessive/limited compute for model inference, what technique should I employ/forgo to get the best out of the budget?” This could be an entry point for enriching quantitative analysis in the paper.

---

> ### Author Response · Authors · 2024-09-19
> **Author response**
>
> ## Author response
>
> Thank you for your review and your feedback, including the positive feedback on our survey’s coverage, selection of works, writing, and presentation.
>
> Please find our responses to your feedback below. We have also integrated the changes that you requested into the paper.
>
> ### Comment 1: (integration of efficient generation)
> > The three areas—generation algorithms, meta-generation, and efficient generation—are presented in parallel, but efficient generation seems to be loosely connected to the first two at first sight: while generation algorithms and meta-generation methods alter the generation process to yield desirable outputs, efficient generation is about reducing cost and improving generation speed. In section 7, the authors touch on how certain efficient techniques could improve aforementioned inference-time methods which evinces its interconnections with the first two topics. I would suggest authors modify the narrative of the abstract and introduction so that efficient generation is not a stand-alone area parallel to the other two, but more like a separate subject that can be applied to the first two areas.
>
> Thank you for this suggestion. We have added a comment about the mutual benefits of developing efficient generation and meta-generation algorithms. We also note that the intro to the efficient generation section emphasizes why meta-generators in particular require efficient generation (for instance, due to calling models multiple times).
>
> ### Comment 2: (quantitative evaluation)
> > It is understandable for a survey paper to be as general as possible to apply to as many cases and scenarios, but it does not hurt to pick a single downstream application/task to offer insights into the uniqueness of each method. For example, authors could elaborate on how certain algorithms behave in NMT tasks.
>
> Thank you for this feedback and suggestion. Several methods in the meta-generation section have been used in the context of mathematical reasoning, so it’s a good domain for showing some quantitative results. We note that Figure 5 shows quantitative results comparing Best-of-N, Majority Voting, and Weighted Voting on the MATH benchmark. We have added a cross-reference to this plot in the meta-generation section.
>
> ### Comment 3: (choose a technique based on the budget)
> > Readers may wonder “When I have excessive/limited compute for model inference, what technique should I employ/forgo to get the best out of the budget?” This could be an entry point for enriching quantitative analysis in the paper.
>
> Thank you for this suggestion. Following the submission of this paper there have been a few papers on choosing inference strategies such as weighted majority voting, MCTS, or other tree search algorithms based on computational budget constraints. We have added a discussion about this in 6.3 (“Minimizing the token budget”). Moreover, we provide token budgets in Table 4 that can be useful for reasoning about which algorithm to choose. However, this is a relatively new area and the studies are done on specific domains (e.g., math or code benchmarks), so specific recommendations about which algorithms to use may evolve as research progresses.
>
> ### Requested:
> > Fig. 2, consider changing candidate tokens to more probable next tokens to simulate real-life cases.
> > Page 24, footnote, mixed usage of notation for matrix transpose and the time dimension.
>
> Thank you for these suggestions! We have adjusted the notation in the revision. We kindly note that the selected tokens in Fig. 2 correspond to lyrics from a popular song. The diagram is designed to show the same 4 tokens in each block, arranged in alphabetical order from top to bottom.
>
> ### Recommended:
> > Section 4.2.3, consider supplying the analytical KL expression for BoN log(n)−n−1n  from Stiennon et al., "Learning to summarize from human feedback", In Proceedings of the 34th International Conference on Neural Information Processing Systems, NIPS’20
> Section 4.3, consider citing Liu et al., "Don't throw away your value model! Generating more preferable text with Value-Guided Monte-Carlo Tree Search decoding", arXiv, 2023
>
> Thank you for these suggestions, we have made them in the revision. In addition to the analytical expression from Stiennon et al, we have also added a comment about [Beirami et al 2024], who show that this expression is an upper bound on the true KL divergence.

---

### Decision · Action_Editor_cdL3 · 2024-10-15

**Recommendation:** Accept as is

**Comment:**

Overall there were only minor concerns with this paper, stemming from notation in a few places and lack of comparison to existing surveys. The authors made updates accordingly and all reviewers recommended acceptance.

**Audience:**

All reviewers agreed the survey would be of interest.

**Claims And Evidence:**

All reviewers felt that this was a comprehensive and clear survey. Since it's a survey, the only "evidence" really comes from explaining/covering past papers in order to provide takeaways and high-level analysis. Overall the reviewers thought the taxonomy, descriptions, characterizations, and takeaways were supported and reasonable. There were some concerns about comparing to existing surveys, but the authors have taken care of this.